# The impact of task context on predicting finger movements in a brain-machine interface

Matthew J Mender[1], Samuel R Nason-Tomaszewski[1], Hisham Temmar[1], Joseph T Costello[2], Dylan M Wallace[3], Matthew S Willsey[1,4], Nishant Ganesh Kumar[5], Theodore A Kung[5], Parag Patil[1,4]*, Cynthia A Chestek[1,3]*

[1]Department of Biomedical Engineering, University of Michigan, Ann Arbor, United States; [2]Department of Electrical Engineering and Computer Science, University of Michigan, Ann Arbor, United States; [3]Department of Robotics, University of Michigan, Ann Arbor, United States; [4]Department of Neurosurgery, University of Michigan Medical School, Ann Arbor, United States; [5]Section of Plastic Surgery, Department of Surgery, University of Michigan, Ann Arbor, United States

**Abstract** A key factor in the clinical translation of brain-machine interfaces (BMIs) for restoring hand motor function will be their robustness to changes in a task. With functional electrical stimulation (FES) for example, the patient's own hand will be used to produce a wide range of forces in otherwise similar movements. To investigate the impact of task changes on BMI performance, we trained two rhesus macaques to control a virtual hand with their physical hand while we added springs to each finger group (index or middle-ring-small) or altered their wrist posture. Using simultaneously recorded intracortical neural activity, finger positions, and electromyography, we found that decoders trained in one context did not generalize well to other contexts, leading to significant increases in prediction error, especially for muscle activations. However, with respect to online BMI control of the virtual hand, changing either the decoder training task context or the hand's physical context during online control had little effect on online performance. We explain this dichotomy by showing that the structure of neural population activity remained similar in new contexts, which could allow for fast adjustment online. Additionally, we found that neural activity shifted trajectories proportional to the required muscle activation in new contexts. This shift in neural activity possibly explains biases to off-context kinematic predictions and suggests a feature that could help predict different magnitude muscle activations while producing similar kinematics.

*For correspondence: pgpatil@med.umich.edu (PP); cchestek@umich.edu (CAC)

Competing interest: The authors declare that no competing interests exist.

## Editor's evaluation

This study provides valuable findings about how brain machine interfaces cope with changes in context, an important consideration for deploying such devices in the real world. The evidence supporting the claims is solid, and the findings will be of interest to motor neuroscientists and engineers developing brain machine interfaces.

## Introduction

Spinal cord injury affects an estimated 296,000 people in the United States (***National Spinal Cord Injury Statistical Center, 2021***). People with quadriplegia have ranked the restoration of hand and arm function as very important for quality of life (***Anderson, 2004***; ***Collinger et al., 2013a***). Functional electrical stimulation (FES) is a therapy that can restore hand and arm function by

electrically stimulating muscles in order to cause contractions. Studies have demonstrated the use of FES to restore at least some hand function since the 1980s (*Kilgore et al., 1989*; *Peckham et al., 1980*), which has resulted in commercially available systems such as the Freehand System (*Peckham et al., 2001*) that was available until the late 2000s. These systems, however, typically relied on external motion or myoelectric commands from residual muscles. These control schemes for FES require residual function and can be unintuitive to use, especially when controlling more than 1-degree-of-freedom.

Brain-machine interfaces (BMIs) have the potential to provide more intuitive control signals that enable people with paralysis to interact with computers, prostheses, or control therapies like FES. These BMIs capabilities have been made possible by a history of neuroscience studies finding that motor cortex activity is correlated with a multitude of movement variables, from intrinsic variables like joint angle and muscle activation (*Evarts, 1968*), to extrinsic variables like movement direction (*Georgopoulos et al., 1986*). Taking advantage of these correlations allows linear models to predict these movement variables from neural activity. This approach has been used in BMIs to allow nonhuman primates to control computer cursors (*Gilja et al., 2012*; *Serruya et al., 2002*; *Taylor et al., 2002*), prosthetic arms (*Carmena et al., 2003*; *Velliste et al., 2008*), and FES (*Badi et al., 2021*; *Ethier et al., 2012*; *Moritz et al., 2008*). Additionally, success in animal BMIs led to the use of similar models in clinical trials as well (*Ajiboye et al., 2017*; *Bouton et al., 2016*; *Collinger et al., 2013b*; *Gilja et al., 2015*; *Wodlinger et al., 2015*). These studies, however, are generally performed in a controlled lab environment, and use relatively simple linear models to make predictions. One key factor in the translation of lab BMI FES systems to tasks of daily living will be how robust they are to the varying environments and tasks found in patient's homes. While some groups have included object interaction in their tasks, for example grabbing single objects (*Ajiboye et al., 2017*; *Downey et al., 2017*), or different size objects (*Wodlinger et al., 2015*), there has not yet been a systematic effort to understand how task context affects BMI performance.

Studies of how motor cortex controls movement have helped to inform how well we can expect BMI models to generalize. This work has shown that the linear encoding of movements in motor cortex can change with many factors such as posture or task duration (*Churchland and Shenoy, 2007*; *Kakei et al., 1999*; *Naufel et al., 2019*; *Scott et al., 2001*; *Sergio et al., 2005*). Recent studies have emphasized instead that the role of motor cortex is to generate movements rather than represent movements (*Churchland et al., 2012*; *Russo et al., 2018*; *Shenoy et al., 2013*). In this view, the activations of single neurons are coordinated. The underlying network connectivity constrains population activity to a low-dimensional manifold and activations on this low-dimensional manifold then form the basis for neural dynamics which generate movements (*Gallego et al., 2017*; *Shenoy et al., 2013*). A key feature of these dynamics that is different from a representation model is that they may have a more computational function, for example ensuring that outgoing commands can be generated reliably (*Russo et al., 2018*). The resulting activity may then change when the same movements are done in different ways because a different computation is needed to generate the movements. As a result, an individual neuron's activity, which is related to the latent activity in this low-dimensional manifold, could correlate with movements differently when the task is changed to one that requires different neural dynamics. With respect to BMI applications, the decoding models assuming a linear relationship, and nonlinear models that do not account for these changes, would then be unable to make accurate predictions in the new tasks.

It is still unclear how large of a task change will require different neural dynamics and thus a different decoding strategy. It has been shown that different dynamics are required for large changes in a task, such as forward versus backward arm pedaling (*Russo et al., 2018*), reaching or walking (*Miri et al., 2017*), or using one arm or the other (*Ames and Churchland, 2019*). At the same time, there is evidence that tasks with the same movements performed differently may have similar neural dynamics. A recent study found that cycling at different speeds led to similar elliptical trajectories in high variance neural dimensions, with a lower variance dimension encoding task speed (*Saxena et al., 2022*). Additionally, a study of isometric, resisted, and free-moving wrist movements found a neural manifold that explained a large amount of neural variance in all tasks (*Gallego et al., 2018*) and a similar study comparing the same wrist movements found that they could still predict muscle activations between the contexts although it required a gain factor related to required muscle activation (*Naufel et al., 2019*). These observations suggest that the neural dynamics may be similar across tasks

with small changes, such as a change in speed or muscle exertion, with differences occurring in lower variance dimensions of population activity.

Which tasks require a change in neural dynamics is a particularly important question to study for hand movements, as the hand is the major end effector interacting with the environment in varying postures and with different loads. However, this work has not yet been extended to continuous finger movements. Finger movements are less studied than arm reaches but initial studies show that grasping movements may show different dynamics due to the increased proprioceptive and tactile feedback present (*Goodman et al., 2019*; *Suresh et al., 2020*). In a promising start to studying decoder generalization for individuated finger movements, it has been shown previously that multiple finger movements can be predicted simultaneously, in real time, and that a linear model trained with data from individual finger movements from two finger groups could also predict combined finger movements (*Nason et al., 2021*), suggesting that individual finger movements and combined finger movements may have similar neural dynamics.

In this study, we investigate how well the decoding of finger movements from intracortical neural activity in nonhuman primates can generalize to realistic alterations of the context in which a task is performed, similar to those that may be found in a BMI user's home. These context shifts represent a small range of the possible shifts but relate back to common musculoskeletal changes in the task, that is muscle length and activation. We ask how the relationship between intracortical neural activity, non-prehensile finger movements, and the corresponding muscle activations are impacted by context changes, such as spring-like resistances and postural changes. We show that these context changes reduce our ability to predict finger kinematics and finger-related muscle activations offline. However, in an online kinematic-based finger BMI task, the monkey can accommodate for the changed task context and achieve near equivalent performance with or without the context change. We explain this by showing that the underlying neural manifolds stay well aligned between contexts, the neural dynamics are shifted due to context, and the shift in neural dynamics can be related to the muscle activation required in the new context.

## Results

### Context changes alter muscle activations and neural activity

We are ultimately interested in understanding the impact of context changes, such as wrist flexion or spring resistance, on BMI decoding performance. In the virtual finger movement task (*Figure 1A*), the monkey moves their fingers within a manipulandum in order to move virtual fingers on a screen in front of them. Cortical spiking activity is recorded during these movements. The monkeys perform center-out and back movements in which they individuate index and middle-ring-small (MRS) finger groups to make one of eight movements (*Figure 1B*) starting from rest, hold the target, then return to rest. In some versions of the task the monkeys performed these movements with all fingers held together for 1-degree-of-freedom (1-DOF, *Figure 1B* bottom). Monkey N additionally had eight chronic electromyography (EMG) leads implanted in muscles of the hand and wrist (see Methods, Table 2) which were recorded from during manipulandum control trials. During the BMI task, a Kalman filter (KF) model is trained to relate cortical activity to finger movements, and the monkey controls the virtual hand with their brain activity through this model. We first asked whether introducing context changes during the manipulandum controlled virtual finger movement task causes any change in behavior, muscle activation, or neural channel activation. Our first manipulations were the addition of torsional springs or the static flexion of the wrist by 23 degrees (*Figure 1C*), referred to as the spring and wrist contexts, respectively, during the 1-DOF center-out task. The torsional springs resist flexion such that more force is required to flex the fingers but less force is required to extend the fingers.

We expected the springs to cause minimal change in finger velocity during movements but a large increase in muscle activation for flexor muscles during flexion and a decrease in activation for extensor muscles during extension. The springs were chosen to be as strong as possible without decreasing the monkey's motivation during the 2-DOF task. As a result, the task could still be completed close to as fast as their reactions allow. *Figure 2A* shows finger position and velocity traces averaged over all flexion trials (solid lines) and all extension trials (dashed lines) on one representative day with the 1-DOF task for Monkey N where the spring manipulation was tested. We see small changes between the velocities in normal trials (black traces) and spring trials (blue traces). To quantify this change, we

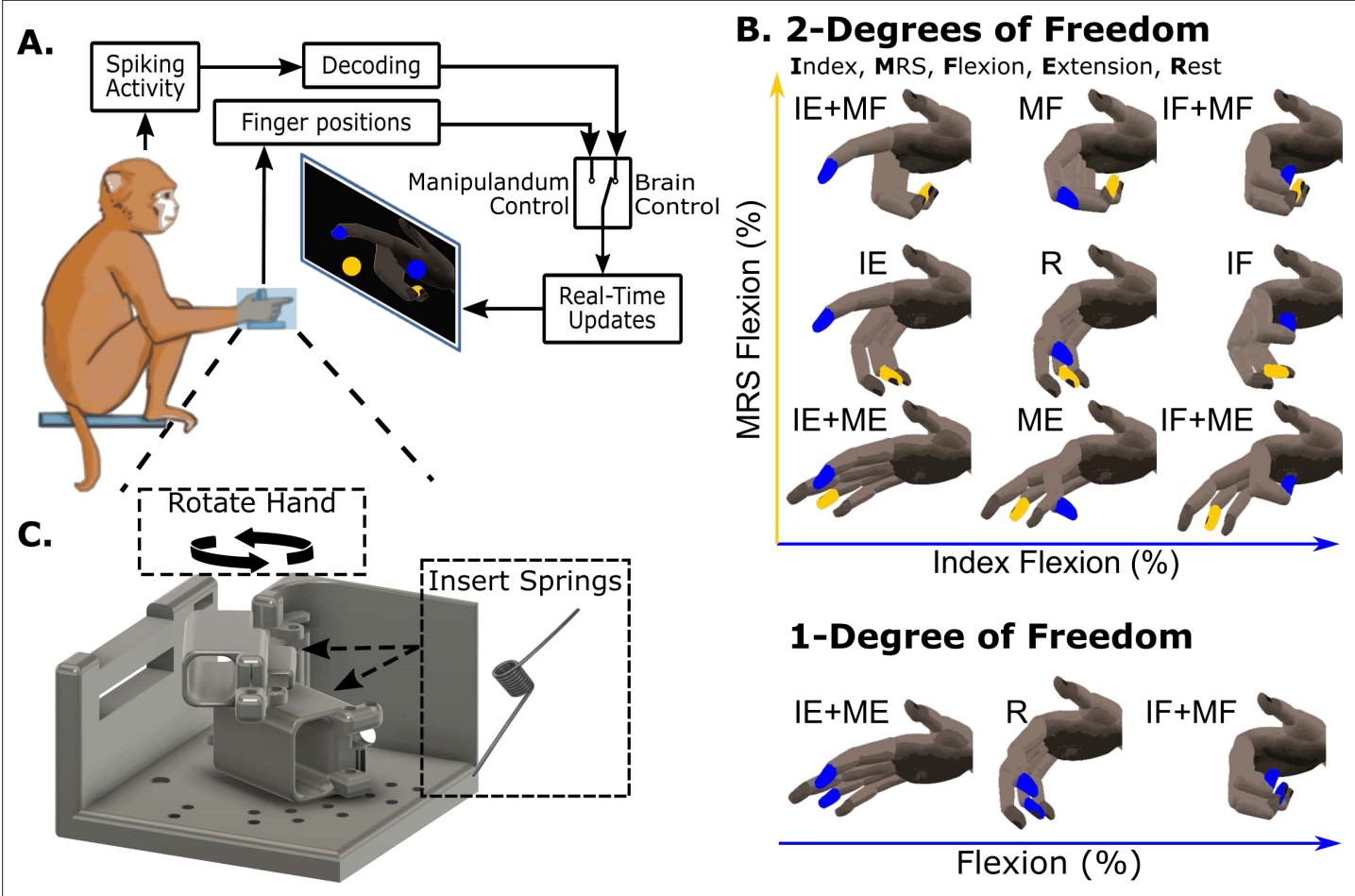

**Figure 1.** Illustration of the behavioral task and context changes. (**A**) Experimental setup during manipulandum control and brain-machine interface (BMI) control experiments. The monkey individuates their index and middle-ring-small finger group, moving each in the manipulandum in order to acquire targets on the screen in front of them. During this task, neural activity and finger positions are both recorded. A model relating neural activity to intended finger movements can be trained and then used in real time to control the virtual hand in front of them. (**B**) Illustration of the possible finger movements. For 2-degree-of-freedom (2-DOF) movements, the index flexion is represented on the x-axis and MRS flexion is represented on the y-axis. In some tasks the monkeys also did a 1-DOF movement, which required flexing or extending all fingers together. (**C**) To alter the context of the task, the manipulandum could be rotated so that the wrist was flexed and torsion springs could be added to the underside of the finger doors. The torsion springs were at rest when the finger doors were at full extension and thus resisted flexion and assisted with extension.

compared the peak velocities between normal trials and other context trials for one representative session with each context (*Figure 2C*). Each monkey had slight behavioral differences in how quickly they performed the task with context changes leading to small changes in peak velocities. We found that the largest changes in peak velocity were Monkey N extending fingers 12.5% faster during wrist trials (p=5e-24, two-sample t-test), and Monkey W flexing fingers 22.3% faster during wrist trials (p=2.6e-11, two-sample t-test), with both monkeys showing small changes in peak movement velocity for at least one movement in each context (p<0.05, two-sample t-test).

In contrast, during the same 1-DOF task, muscle activations change substantially for trials toward both targets (*Figure 2B*), showing trends such as increased flexor digitorum profundus (FDP) muscle activation for flexion and less extensor digitorum communis (EDC) activation for extension. All muscles implanted for Monkey N are included in Table 2 (Methods) while Monkey W did not have EMG electrodes. Using the same representative sessions for Monkey N, we compared the average muscle activations from Monkey N in a 420ms window around peak movement between normal trials and off-context trials (*Figure 2D, E*). During spring trials, we found that every muscle except FCR required significantly higher than normal muscle activation for flexion (*Figure 2D* blue, p<0.004 two-sample t-test), an average increase of 91.9% for the finger flexor muscles (FDPid, FDPip, FDP), and every

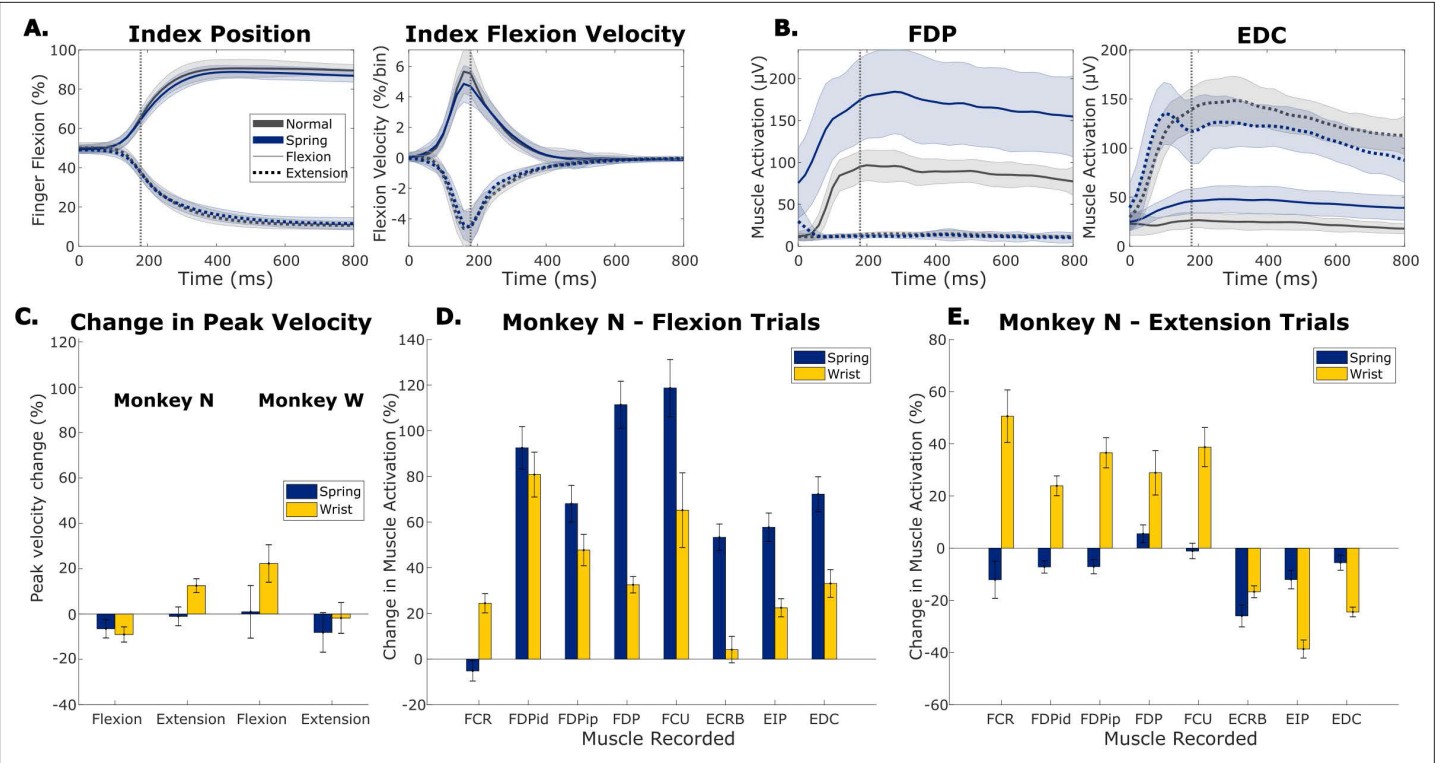

**Figure 2.** The impact of context changes on kinematics and muscle activation. (**A**) Trial-averaged traces of index finger position and index finger flexion velocity for an example 1-degree-of-freedom (1-DOF) spring session with Monkey N. Trials are aligned to peak movement (vertical gray line). Black traces are normal trials, blue are trials with springs in the manipulandum, solid traces are for flexion trials, and dotted traces are for extension trials. Shaded area shows one standard deviation. (**B**) Trial-averaged traces of flexor digitorum profundus (FDP) muscle activation and extensor digitorum communis (EDC) muscle activation for an example spring session with Monkey N. Formatted the same as (**A**). (**C**) Change in peak velocity between normal trials and trials with either springs present or the wrist flexed for both Monkey N (left) and Monkey W (right). Trials are split by movement direction, either flexion or extension. Error bars indicate 99% confidence interval based on a two-sample t-test. (**D, E**) Change in average muscle activation in a window around peak movement between normal trials and trials in the spring (blue) or wrist context (yellow) for all eight muscles recorded in Monkey N. Trials are split between flexion (**D**) and extension (**E**) movements. Error bars indicate 99% confidence interval based on a two-sample t-test.

muscle except FDP and FCU required less muscle activation for extension (**Figure 2E** blue, p<1e-5 two-sample t-test), an average decrease of 8.2% in finger extensor muscles (EIP, EDC). Interestingly, even extensor muscles were more activated during spring flexion trials, indicating that Monkey N was co-contracting muscles more and moving with more stiffness. During wrist trials, finger flexor muscles showed an average 53.3% increase in activation for flexion trials (**Figure 2D** yellow) and finger extensor muscles had an average 32% decrease in activation for extension trials (**Figure 2E** yellow).

After establishing that context had a large effect on muscle activity with a relatively small effect on finger kinematics, we next evaluated whether neural activity changed due to the addition of springs or altered wrist posture. For each neural channel we recorded two features, the threshold crossing firing rate (TCFR) and spiking band power (SBP). SBP is a low-power feature that has been previously shown to be well correlated with the firing rate of the largest amplitude unit (**Nason et al., 2020**), often enabling us to identify more tuned channels. For both Monkey N and Monkey W, we evaluated how many channels were tuned to movement and how many of these tuned channels modulated activity with context change. Tuning and context modulation were determined by regressing finger kinematics with channel activity and channel activity multiplied by a dummy variable for context, as described in the Methods, one channel at a time. Regression coefficients were tested for significance with a t-test, a significant channel activity coefficient indicated that channel was tuned, and a significant dummy variable coefficient indicated that context modulated the channel's tuning. The results are included in **Table 1**. The SBP feature resulted in an average of 86.9 and 28 tuned channels of 96 for Monkey N and Monkey W, respectively, while TCFR resulted in an average of 36.7 and 11.8 tuned channels of 96 for Monkey N and Monkey W, respectively. An average of 24.4% of the tuned TCFR

**Table 1.** The number of channels tuned to any movement using two features (threshold crossing firing rate [TCFR] and spiking band power [SBP]) and the percentage of tuned channels that showed a significant change in neural feature between normal trials and trials in the tested context for four types of experimental sessions (Monkey N or Monkey W with the spring or wrist context).
Standard deviation (SD) is calculated across sessions of the same type and n is the number of sessions of that type which is indicated in the first column.

| | TCFR | | SBP | |
| --- | --- | --- | --- | --- |
| | Tuned channels | % Context modulated channels | Tuned channels | % Context modulated channels |
| Monkey N spring days (n=3) | 38.3 (SD = 4.0) | 47.0% (SD = 2.8) | 89.3 (SD = 2.1) | 61.7% (SD = 12.0) |
| Monkey N wrist days (n=2) | 35.0 (SD = 1.4) | 43.1% (SD = 9.8) | 84.5 (SD = 7.8) | 54.4% (SD = 0.8) |
| Monkey W spring days (n=3) | 12.0 (SD = 1.0) | 66.5% (SD = 2.8) | 35.0 (SD = 6.9) | 42.8% (SD = 7.1) |
| Monkey W wrist days (n=2) | 11.5 (SD = 3.5) | 5.6% (SD = 7.9) | 21.0 (SD = 5.7) | 20.9% (SD = 4.5) |

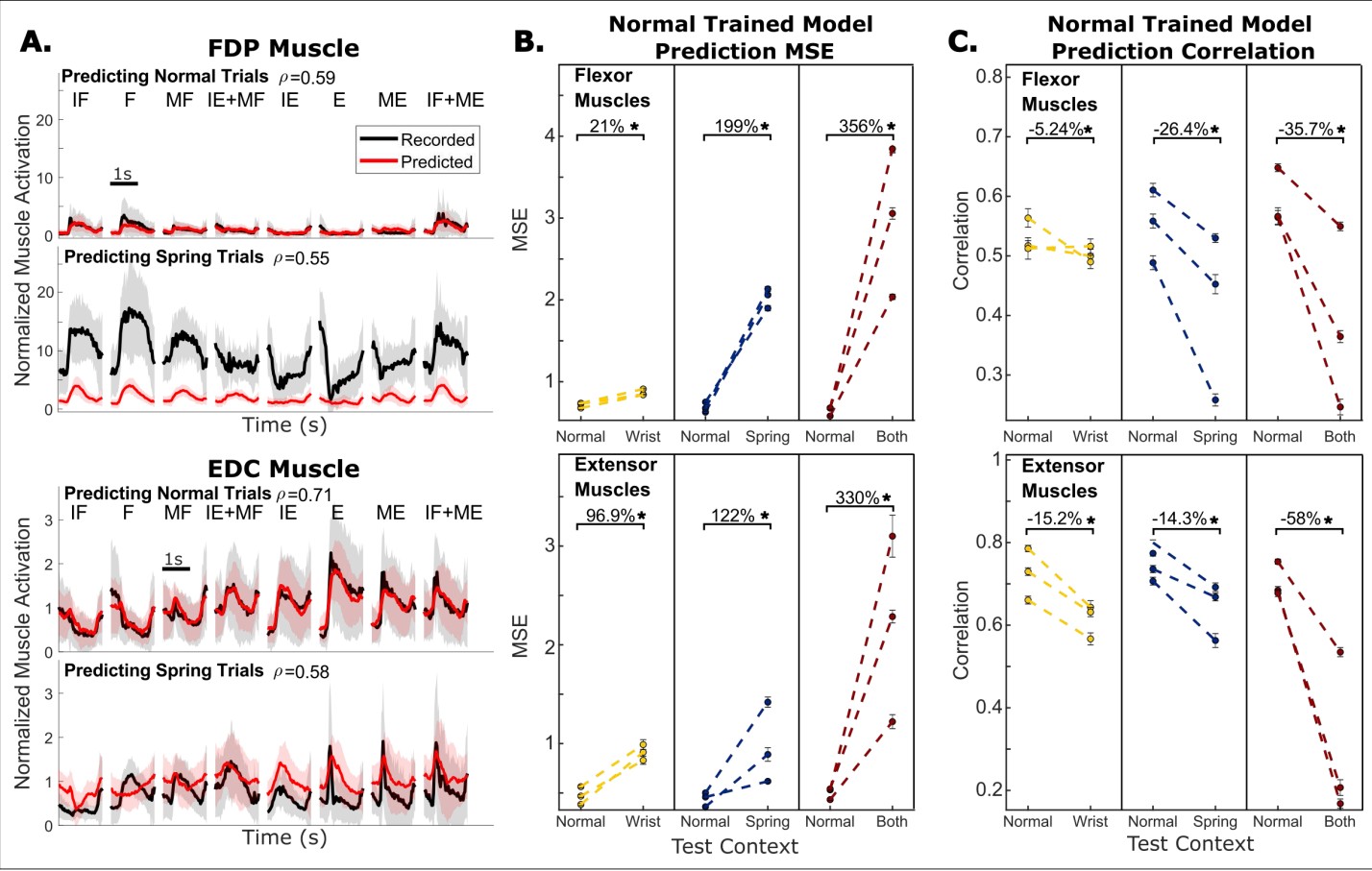

**Figure 3.** Offline predictions of muscle activations. (**A**) Recorded and predicted muscle activation traces for flexor digitorum profundus (FDP) muscle (top half) and extensor digitorum communis (EDC) muscle (bottom half) from one example session with Monkey N. Traces are aligned to peak movement and averaged over trials to the same target, shading represents one standard deviation. Predictions are from a model trained only on normal trials and the model is evaluated either on normal trials (top) or trials with springs present (bottom). r indicates the linear correlation coefficient between recorded and predicted muscle activations using all trials in the specified context for that session, excluding the normal context trials used for training the model where applicable. IF – index flexion, MF – MRS flexion, F – both fingers flexion, IE – index extension, ME – MRS extension, (**E**) – both fingers extension. (**B**) Change in prediction mean-squared error (MSE) when a model trained on normal trials is evaluated on trials in a different context. Color indicates which context is being tested, yellow is wrist, blue is spring, and both is red. Each dashed line and pair of dots represent one session during which the same model was used for both measurements. Error bars on the dots indicate one standard deviation for model performance calculated with 10-fold cross-validation. (**C**) Same as (**B**) but model performance is measured with prediction correlation.

channels and 37.7% of the tuned SBP channels significantly changed activity with the wrist context and an average of 56.8% of the tuned TCFR channels and 52.3% of the tuned SBP channels significantly changed activity with the spring context. As both features had a similar proportion of tuned channels that were modulated by context changes, we opted to use SBP as the primary feature for the subsequent analyses in order to increase the number of tuned channels available for analysis.

## Decoding neural activity across task context

After confirming that these context changes had large impacts on muscle activation (*Figure 2*) and affected many channels of neural activity (*Table 1*), we next asked how this will impact the ability to decode intended movements for BMI applications. Typically, BMIs use linear models to relate neural firing rates to the desired control variable (*Ajiboye et al., 2017*; *Nason et al., 2021*; *Wodlinger et al., 2015*). Given the work showing that task changes similar to those tested here can alter how motor cortex linearly encodes muscle activations during different wrist movements across tasks (*Naufel et al., 2019*), we next ask if the same is true for individuated finger movements. To test this, we recorded kinematics for both monkeys and muscle activations for Monkey N during the 2-DOF task

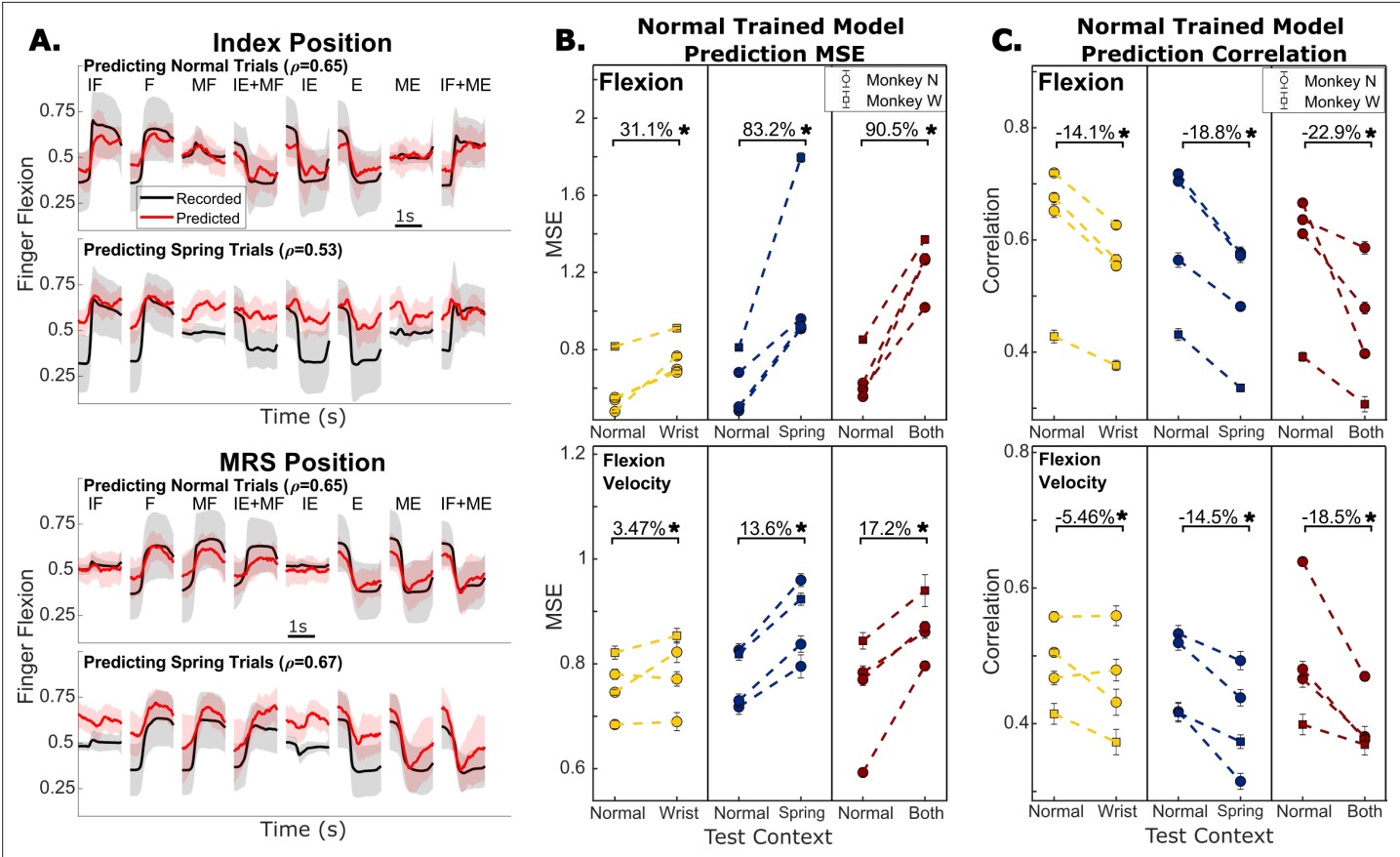

**Figure 4.** Offline predictions of kinematics. (**A**) Recorded and predicted position traces for index finger position (top) and middle-ring-small (MRS) finger group position (bottom), averaged across all trials toward each target. Shading represents one standard deviation. Predictions are from a model trained only on normal trials and the model is evaluated either on normal trials or trials with springs present. r indicates the linear correlation coefficient between recorded and predicted finger positions using all trials in the specified context for that session, excluding the normal context trials used for training the model where applicable. (**B**) Change in prediction mean-squared error (MSE) when a model trained on normal trials is evaluated on trials in a different context. Color indicates which context is being tested, yellow is wrist, blue is spring, and both is red. Each dashed line and pair of dots represent one session during which the same model was used for both measurements. Error bars on the dots indicate one standard deviation for model performance calculated with 10-fold cross-validation. (**C**) Same as (**B**) but model performance is measured with prediction correlation.

and then trained linear models with data from normal trials to predict muscle activations or kinematics in unseen normal trials or other context trials.

We first present the results for decoding Monkey N's muscle activations across context. *Figure 3A* shows average predictions of FDP and EDC muscle activations for normal trials and spring context trials from one example experimental day, both using a linear model trained on normal trials. We found that within-context linear models, that is models trained and tested on trials of the same context, could predict muscle activations well during individuated finger movements, with accuracy comparable to predictions of kinematics (*Supplementary files 1 and 2*). However, models trained on normal trials are consistently unable to predict muscle activations well in the off-context trials (*Figure 3B*). For example, when springs are present the predictions do not account for the large changes in FDP activation magnitude or EDC activation during flexion trials (*Figure 3A*, *Figure 4*). Across three sessions in each context including the wrist, spring, or both wrist and spring contexts, prediction mean-squared error (MSE) increased significantly from the normal trial baseline. This held true for both flexor muscles (FDP, FDPip) and extensor muscles (EDC, EIP) (evaluated by paired t-test, $p<2e-9$), in each tested context, with an average increase of 188.7% across all context changes and muscles. The increases in error varied widely, ranging from a 21% increase (flexor muscles with wrist-flexed) to a 356% increase (flexor muscles with both wrist-flexed and springs).

We next asked whether this dramatic increase in prediction error is driven by a simple offset or magnitude change, or a reduced linear relationship with recorded muscle activations. For example, while the off-context predictions of FDP activation during flexion in *Figure 3A* both do not account for the offset in muscle activation at the beginning of trials and do not predict a large enough change in the magnitude of muscle activation throughout the trial, the same linear correlation is maintained. As a result, these predictions might only need a bias and scaling adjustment to recover performance. Alternatively, during flexion, the off-context predictions of EDC activation are less correlated with measured EDC activation because the model predicts EDC inactivation, which occurs during normal trials. However, in spring trials, measured EDC activation actually increases for flexion due to Monkey N co-contracting to move more stiffly. Since MSE is influenced by both changes to linear correlation and changes to offsets and scaling, we also measured prediction correlation (*Figure 3C*) which is less affected by the changes to offsets and scaling. Using the same sessions and models trained on normal trials as when we measured MSE, we found that prediction correlation decreased from normal baseline by an average of 25.8% across tested contexts and muscles. This change was significant for all tested contexts (paired t-test), ranging from a 2.4% decrease for FDPip in the wrist context (p=0.03) to a 69.4% decrease for EDC in the both context (p=1.3e-17). While significant, the change in correlation was a smaller effect than the change in MSE.

Kinematics are used as a control signal in BMI applications more frequently than muscle activation so we next examined the error in predicting finger position and velocity across contexts. *Figure 4A* shows trial-averaged predictions for each target from training a linear model on normal trials and predicting normal trials or spring trials for an example session with Monkey N. In both index and MRS flexion predictions, we observed the off-context predictions to be worse than the normal trial predictions. Predictions during the spring trials often showed a bias towards flexion. We measured changes in prediction accuracies on three days for each context – spring, wrist, and both – for Monkey N, and one additional day with each context for Monkey W (*Figure 4B*, *Figure 4C*). All context changes resulted in significantly higher prediction MSE (paired t-test, p<1e-4), averaging 68.2% for finger position and 11.4% for finger velocity. All context changes also resulted in small but significant decreases in prediction correlation (paired t-test, p<1e-4), averaging -18.6% for finger position and -12.8% for finger velocity. The smaller change in the correlation of position predictions indicates that much of the prediction error is coming from offsets or magnitude differences in the predictions.

## Changing task context has small effects on online BMI performance

Based on these offline prediction results, we might expect that in a real-time BMI when cortical activity is controlling the virtual hand, a model trained on normal trials will be more difficult to use when controlling a virtual hand in a new context. We investigated this by training either a KF or a ReFIT Kalman filter (RFKF), as done previously by *Nason et al., 2021*, and having the monkey control the virtual hand with the model while we applied context changes to this virtual task. Briefly, the KFs are standard position/velocity KFs that update virtual finger position by integrating the predicted finger velocity in the current time step. We introduced context changes in two separate ways. First, we added springs, a static wrist flexion, or both to the manipulandum and had the monkey control the virtual hand with an RFKF trained on normal trials. Second, we trained different KFs using training data collected in different contexts and had the monkeys use the KFs in the online task without any context changes applied to the manipulandum. Due to the quality of recorded neural signals, Monkey W controlled only 1-DOF online while Monkey N controlled 2-DOF online.

We first tested whether online BMI performance changed when using a standard RFKF with context changes added to the manipulandum, referred to as the manipulandum context change BMI experiments. One RFKF model could be tested on multiple context changes in a single session. For example, *Figure 5A* shows the acquisition times during an experimental session where two contexts, spring and wrist, were tested in sets of separate trials. *Figure 5B* summarizes the changes in online performance over six experimental sessions for Monkey N and four sessions for Monkey W. During these 10 sessions the context changes were tested 15 times: four times for the wrist context, seven times for the spring context, and four times for the combined wrist and spring context. Each bar compares the performance during one of the 15 tests between normal trials and one off-context condition in a session when using the same model for both. In two of these tests with Monkey N (one spring and one combined wrist and spring), random target presentation was used instead of center-out to

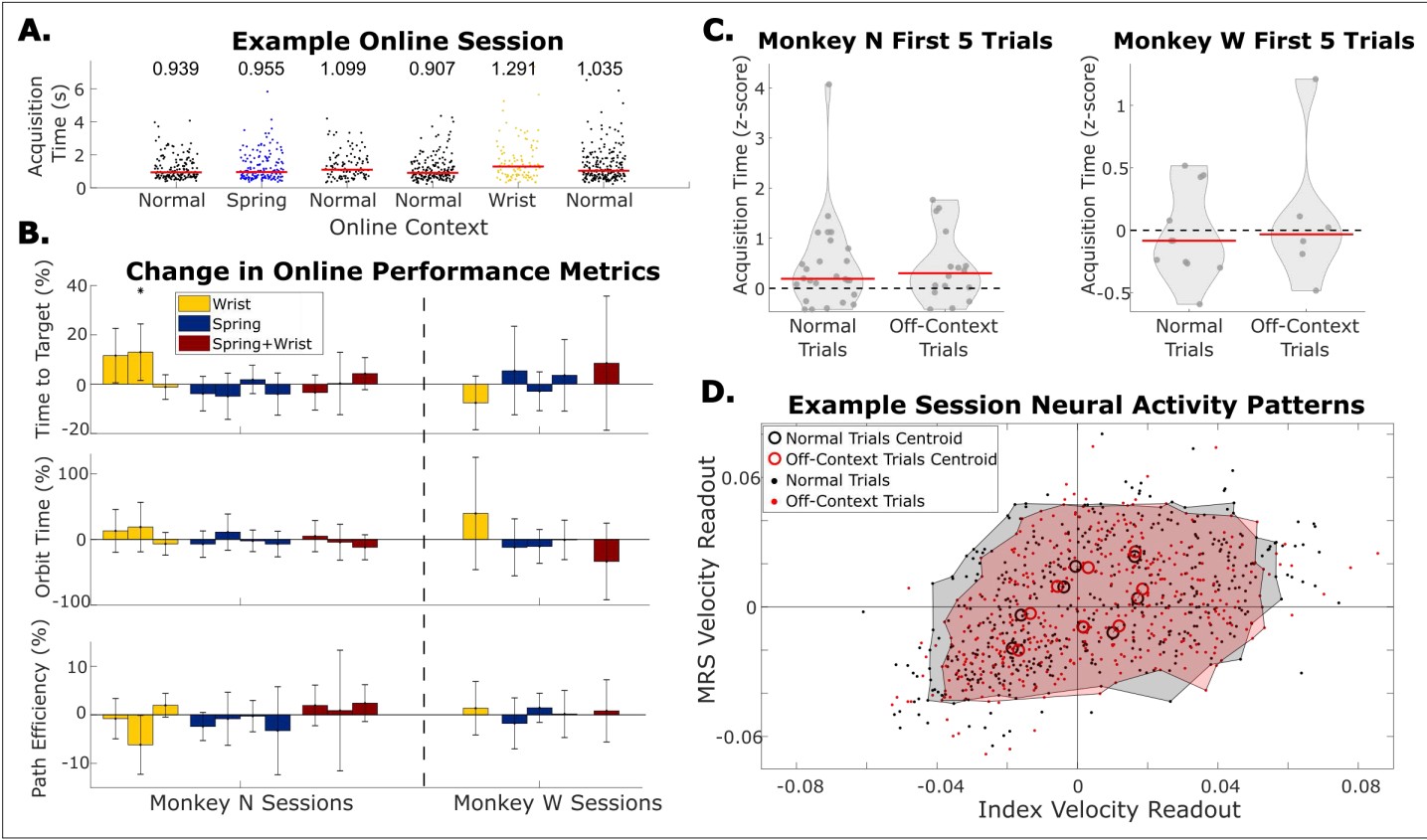

**Figure 5.** Online performance when context changes are tested by adding changes to the manipulandum during online trials. (**A**) Example online session in which both the spring and the wrist context are tested. Each dot indicates acquisition time of one trial, each grouping of dots is a series of trials before the context was changed. Red bars and numbers above each grouping illustrate the median acquisition time (in seconds) for that series of trials. (**B**) Change in performance metrics between normal online trials and online trials with the context change indicated by the bar color in the manipulandum. Each bar indicates one session where off-context online trials are compared to the normal online trials immediately before and after them. Error bars indicate 99% confidence interval in performance metric change. Dashed line separates Monkey N sessions from Monkey W sessions. (**C**) Average acquisition time during the first five trials each time online trials were started, split between normal trials and trials with context changes applied to the hand (off-context). Acquisition times were z-scored within a series of trials in the same context. Red lines indicate the median. (**D**) Neural activity patterns for one example session. Neural activity patterns are velocity predictions at the time point of peak brain-machine interface (BMI) movement using a single linear regression model trained on normal offline trials. Each dot indicates the readout velocity for one trial using the same linear model but for either a normal trial (black) or an off-context trial (red). Larger open circles indicate the centroid of velocity readouts for trials to one of eight target directions split by normal and off-context trials. Shaded areas bound patterns for all trials excluding trials outside the 95th percentile of index or middle-ring-small (MRS) velocities.

The online version of this article includes the following figure supplement(s) for figure 5:

**Figure supplement 1.** The correlation between hand position and online decode during the trials for each online comparison in *Figure 5B*.

**Figure supplement 2.** Change in 'pushing' magnitude, that is the predicted velocity along the target direction, with predictions made by a linear regression model.

increase task difficulty. Ultimately, both monkeys reached the same levels of performance despite added context changes to the manipulandum. Of the 15 tests, only one test resulted in a significant change in at least one of the performance metrics (p<0.01, two-sample t-test). In this case, Monkey N using the RFKF while his wrist was flexed resulted in a 13.0% increase in time to target (p=6.7e-3), the equivalent of 86 ms. This overall lack of change was somewhat surprising since the offline decoding results had greater prediction error. The expectation was that when the monkey moved their hand along with the BMI task, the performance would be impacted due to the context change. However, the data show that the monkeys made small adjustments to how their hand moved with the online task (*Figure 5—figure supplement 1*).

To measure the amount that the monkeys had to adjust during online trials to get to average performance, we calculated the average acquisition time, defined as the time to reach the target plus

the time to finish orbiting the target, for the first five trials after the start of online trials and compared that between normal and off-context runs of BMI trials. Acquisition times were z-scored within a series of trials performed in the same context before calculating the average in the first five trials. *Figure 5C* shows the distribution of these average acquisition times for every instance the online trials were started, split between normal trials and off-context trials. Monkey N had slightly worse initial performance during normal online BMI use as the average acquisition time during the first five trials was significantly greater than zero (p=0.002, one-sample Kolmogorov-Smirnov test). Monkey W, on the other hand, did not have significant adaptation from the first five BMI trials (p=0.22, one-sample Kolmogorov-Smirnov test). Interestingly, the performance in the first five off-context trials is not different from normal trials for both monkeys (two-sample Kolmogorov-Smirnov test, p=0.88 for Monkey N, p=0.79 for Monkey W). This suggests that adaptation to BMI with the context changes tested here is as difficult as adaptation from hand control to BMI control.

To help explain this minimal change in online performance, we examined the monkeys' neural activity during online trials. Similar to other BMI adaptation work (*Golub et al., 2018*), we used predicted velocities as a low-dimensional behaviorally relevant readout of neural activity during online trials. The velocity predictions are made using a linear regression model trained on SBP from the normal offline training trials in that session with an additional 250 ms history (five 50 ms bins) of SBP from each channel appended as additional features. We call the predicted index and MRS velocity at the time of peak online velocity the 'neural activity pattern' for that trial. An example session for Monkey N is shown in *Figure 5D* with trials split between normal BMI trials (black dots) and off-context BMI trials (red dots). This session was one of two sessions near the median change in online acquisition time between contexts across the 10 sessions for Monkey N. Open circles show the centroid of velocity readouts for all trials in that context toward the same target. The close proximity of the centroids and the overlap of the cluster of normal and off-context points in general indicate that similar neural activity patterns were being produced. This suggests that the monkey was using the same strategy in both types of trials even though the precise patterns may have differed. Across 13 online tests, the target neural activity pattern centroids did not change their magnitude along the target direction when compared between context for flexion targets, extension targets, and split targets (*Figure 5—figure supplement 2*, two-sample t-test with 5% false discovery rate correction). Note that we excluded the two sessions with random target presentation from this analysis because trials did not have consistent target directions for calculating centroids.

In a second online experiment, referred to as the two decoder BMI experiments, the monkeys alternated between using two KFs: one trained on normal trials and another trained on off-context trials. In this paradigm the context change is added to the model used in closed loop BMI so that it directly impacts BMI control. Monkeys N and W performed these tests on 9 and 6 separate days, respectively. On each day, two decoders were trained in order to compare one context change. *Figure 6A* shows an example session alternating between a decoder trained on normal trials and a decoder trained on wrist trials. *Figure 6B* shows the performance changes for 15 sessions with five, six, and four sessions testing the wrist, spring, and combined wrist and spring contexts, respectively. In 15 sessions, this experiment revealed small but significant changes in at least one performance metric for 11 sessions (*Figure 6B*, p<0.01, two-sample t-test) although only two sessions had worse online performance for all three metrics. The significant decreases in performance averaged 32.6% for time to target, 46.5% for orbit time, and 8.5% for path efficiency, with the combined context having the largest effect.

We compared the acquisition time in the first five trials while using the normal decoder or an off-context decoder (*Figure 6C*). Normal and off-context trials on average did not show different relative performance in the first five trials. Similar to *Figure 5C*, Monkey N had higher acquisition times in the first five trials (p=0.008, one-sample Kolmogorov-Smirnov test) that is the same between using the normal model and off-context models (p=0.99, two-sample Kolmogorov-Smirnov test). Monkey W once again did not show a significant initial adaptation (p=0.17, one-sample Kolmogorov-Smirnov test) which was the same between using the normal model and off-context models (p=0.5, two-sample Kolmogorov-Smirnov test). This indicates that for both monkeys, adapting to the off-context decoder was as difficult as adapting to the normal decoder.

As the off-context online performance was worse in many of the two decoder BMI sessions, we next asked if this BMI task required more adaptation than when context changes were added to the manipulandum. As done previously, we calculated neural activity patterns, that is velocity readouts

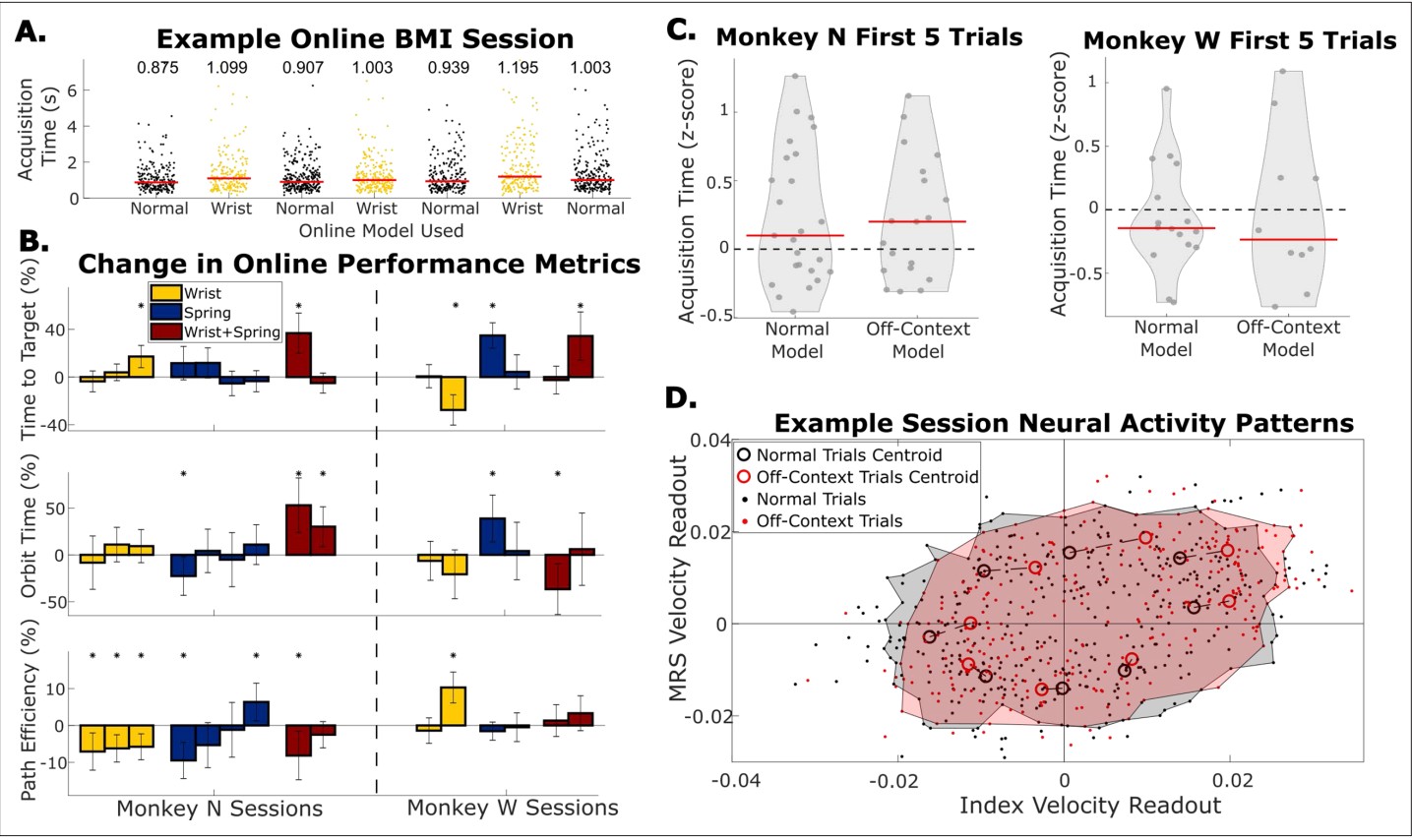

**Figure 6.** Online performance when context changes were tested by using decoders trained with normal training data or off-context training data. (**A**) Example online session in which the wrist context was tested. Each dot indicates acquisition time of one trial, and each grouping of dots is a series of trials before the context was changed. Red bars and numbers above each grouping illustrate the median acquisition time (in seconds) for that series of trials. (**B**) Change in performance metrics between normal online trials and online trials with the context change indicated by the bar color. Each bar represents one session where off-context online trials are compared to the normal online trials immediately before and after them. Error bars indicate 99% confidence interval in performance metric change. The dashed line separates Monkey N sessions from Monkey W sessions. (**C**) Average acquisition time during the first five trials each time online trials were started, split between trials performed with the model trained on normal trials and the model trained on off-context trials. Acquisition times were z-scored within a series of trials with the same model. Red lines indicate the median. (**D**) Neural activity patterns for one example session. Neural activity patterns are velocity predictions at the time point of peak brain-machine interface (BMI) movement using a single linear regression model trained on normal offline trials. Each dot indicates the readout velocity for one trial using the same linear model but for either a normal trial (black) or an off-context trial (red). Larger open circles indicate the centroid of velocity readouts for trials to one of eight target directions split by normal vs off-context trials. Shaded areas bound patterns for all trials excluding trials outside the 95th percentile of index or middle-ring-small (MRS) velocities.

The online version of this article includes the following figure supplement(s) for figure 6:

**Figure supplement 1.** Correlation between the velocity predictions made with each Kalman filter used in the two-model brain-machine interface (BMI) experiments for Monkey N.

**Figure supplement 2.** Change in 'pushing' magnitude, that is the predicted velocity along the target direction, with predictions made by a linear regression model.

from neural activity for each online trial. These patterns were calculated using one linear regression model trained on the normal context offline training trials from the same session. Ultimately, observed adaptation was a small effect, likely due to very high correlations between the velocity decodes with both KFs (*Figure 6—figure supplement 1*). Neural activity patterns for an example session for Monkey N with the median change in acquisition time are shown in *Figure 6D*. Neural activity patterns for trials using the normal model are represented in black and patterns for trials using the off-context model are represented in red. While the overall repertoire of neural activation patterns largely overlaps, we saw small shifts in the centroids of patterns for individual targets. These shifts in this session included higher velocity for flexion and smaller velocities for extension in the off-context trials. A shift

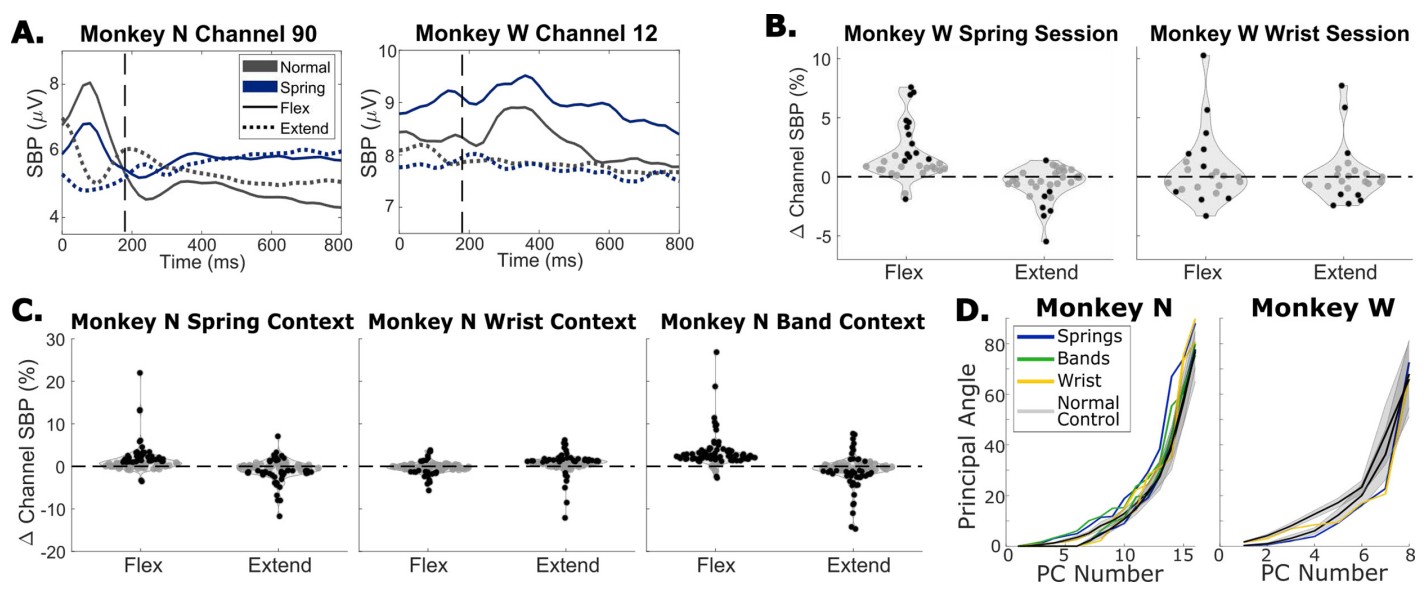

**Figure 7.** The impact of context changes on neural activity. (**A**) Trial-averaged spiking band power (SBP) for two channels during a spring session for each monkey. Trials were aligned to peak movement before averaging, vertical dashed lines indicate peak movement. Black traces show normal trials, blue traces show trials with springs present, solid traces are flexion trials, and dotted traces are extension trials. (**B**) Change in average SBP in a window around peak movement for each tuned channel for a spring session (left) and wrist session (right), both for Monkey W. Black dots indicate significant differences according to a two-sample t-test (p<0.01). (**C**) Same as (**B**) but for Monkey N and using one session where the task was performed normally and in the spring, wrist, and rubber band contexts. (**D**) Principal angles between the PCA space calculated for normal trials and the PCA spaces calculated for spring trials (blue), band trials (green), or wrist trials (yellow). Black lines are average angles between PCA spaces for two random sets of normal trials, with gray shading indicating one standard deviation.

toward higher velocities suggests that the monkey was 'pushing' harder during those trials. When comparing these centroids in all sessions the shifts along the target direction were generally larger for these two decoder sessions than the manipulandum context changing sessions (*Figure 5—figure supplement 2*, *Figure 6—figure supplement 2*). Additionally, across the 15 sessions there was a trend that if the monkey had to push harder, that would happen during flexion trials (*Figure 6—figure supplement 2*), all three significant increases (two-sample t-test with 5% false discovery rate correction) were for flexion targets. Altogether, this indicates a small trend that the monkeys would re-aim during off-context flexion trials in the two decoder sessions by aiming for a target further from center (i.e. pushing harder).

## Context shifts population neural activity

To help explain how the monkeys were able to adjust to different contexts during the online task, we further examined changes in neural activity during the offline task in different contexts. First, we ask if there are any obvious trends in how the channel activity changes during simple 1-DOF movements, for example increasing neural activation when flexion requires more muscle activation. In one experimental session, Monkey N performed the 1-DOF task normally as well as in the wrist, spring, and rubber band contexts. The rubber bands altered the required muscle activations for the task in the same way as the springs, however to a larger extent, and as such were only used in this 1-DOF task. In two additional sessions, Monkey W performed the 1-DOF task normally as well as in the wrist context in one session and spring context in the other session. *Figure 7A* shows trial-averaged neural activation traces from two example modulated channels, one from each monkey, both comparing the activation during spring trials and normal trials. We found that neural channels showed a mix of changes with context. For example, Monkey W's channel 12 was activated more compared to normal for spring flexion targets (blue solid), similar to the muscle activations of the finger flexors. However, other example channels like Monkey N's channel 90 show less neural activation during movement in the spring contexts for both flexion and extension targets.

To quantify changes in activation for the population of tuned channels, we compared the average channel activation in a window spanning 420 ms around peak movement for each type of trial. *Figure 7B and C* shows the change in SBP for all tuned channels between off-context trials and normal trials, split by flexion and extension trials, for Monkey W and Monkey N, respectively. Black dots indicate channels with significantly different trial SBP between off-context and normal trials toward that target according to a two-sample t-test (p<0.01). During spring trials, context modulated channels were activated significantly less on average for extension and were activated significantly more for flexion with both monkeys (p<0.01, paired t-test). During wrist trials, context modulated channels were activated significantly less for extension for Monkey N only (p<0.01, paired t-test) and there were no trends for channels increasing or decreasing activation on average for flexion or extension for Monkey W. Notably, the majority of changes in neural activation are on the order of 10% or less for individual channels, fairly small relative to the large changes in muscle activation observed in *Figure 2*, which is consistent with the results from *Naufel et al., 2019*, with wrist movements.

We next investigated how consistent the covariance structure of the neural activity is across different task contexts. We calculated the principal components (PCs) underlying the neural activity in each context in order to obtain one manifold for each context. We then found the minimum angles, also known as the principal angles, required to align the PCs from each type of off-context trial with the PCs calculated from normal trials (*Figure 7D*), similar to what has been previously presented (*Gallego et al., 2018*). Principal angles were also calculated between manifolds calculated from random sets of normal trials to create a set of control angles. Two sessions with normal, spring, wrist flexed, and rubber band trials in the same session are included for Monkey N (*Figure 7D* Left), and two sessions, one with normal and wrist flexed and one with normal and spring trials, are included for Monkey W (*Figure 7D* Right). We found that the principal angles between off-context trial neural activity and normal trial neural activity match the normal control angles well. This indicates that the activity in each context falls within well-aligned manifolds.

Next we looked at how much variance in neural activity is due to the context changes. We calculated 16-dimensional demixed PCA (dPCA, *Kobak et al., 2016*) components for a neural manifold spanning neural activity during trials from all contexts in a single session. *Figure 8A* shows the dPCA components for one session with Monkey N. The components are organized in rows according to which behavioral parameter they explain the most variance for: time (condition-independent), context, target, or context-target interaction. Using the four sessions included in *Figure 7D* and one additional session for Monkey N that included normal, spring, and rubber band trials, the amount of variance explained by each behavioral parameter is summarized in *Figure 8B* for both monkeys. The condition-independent and target components together explain the majority of the neural variance. On average, the target components explain 36.4% of neural variance, the condition-independent components explain 47.2% of neural variance, and the context and context-target interaction components together explain 24.1% of neural variance for Monkey N and 8.6% of neural variance for Monkey W.

In inspecting the activation of components in *Figure 8A*, the context-related components add a shift to neural activity before and after movement (component #3) and separate normal and wrist trials from spring and rubber band trials during movement (component #6). This creates the picture of trajectories that are largely the same between context but slightly shifted, perhaps in response to a change in proprioceptive input or to generate more muscle activation. We compared the average activation around peak movement of the context-dependent dPCA component that explained the most neural variance with the average muscle activation of flexor or extensor muscles during flexion or extension trials in each context. *Figure 8C* shows this comparison for the three sessions with Monkey N included in *Figure 8B*. Note that one session only included normal, spring, and rubber band trials (no wrist). Interestingly, we found that the first context-dependent component correlated very strongly with the activation of the active muscle (i.e. flexor muscles during flexion or extensor muscles during extension) across contexts, with correlations of 0.91 for flexor muscles and –0.89 for extensor muscles. This result may suggest that a feature from neural activity could be used to account for changes in required muscle activation between contexts.

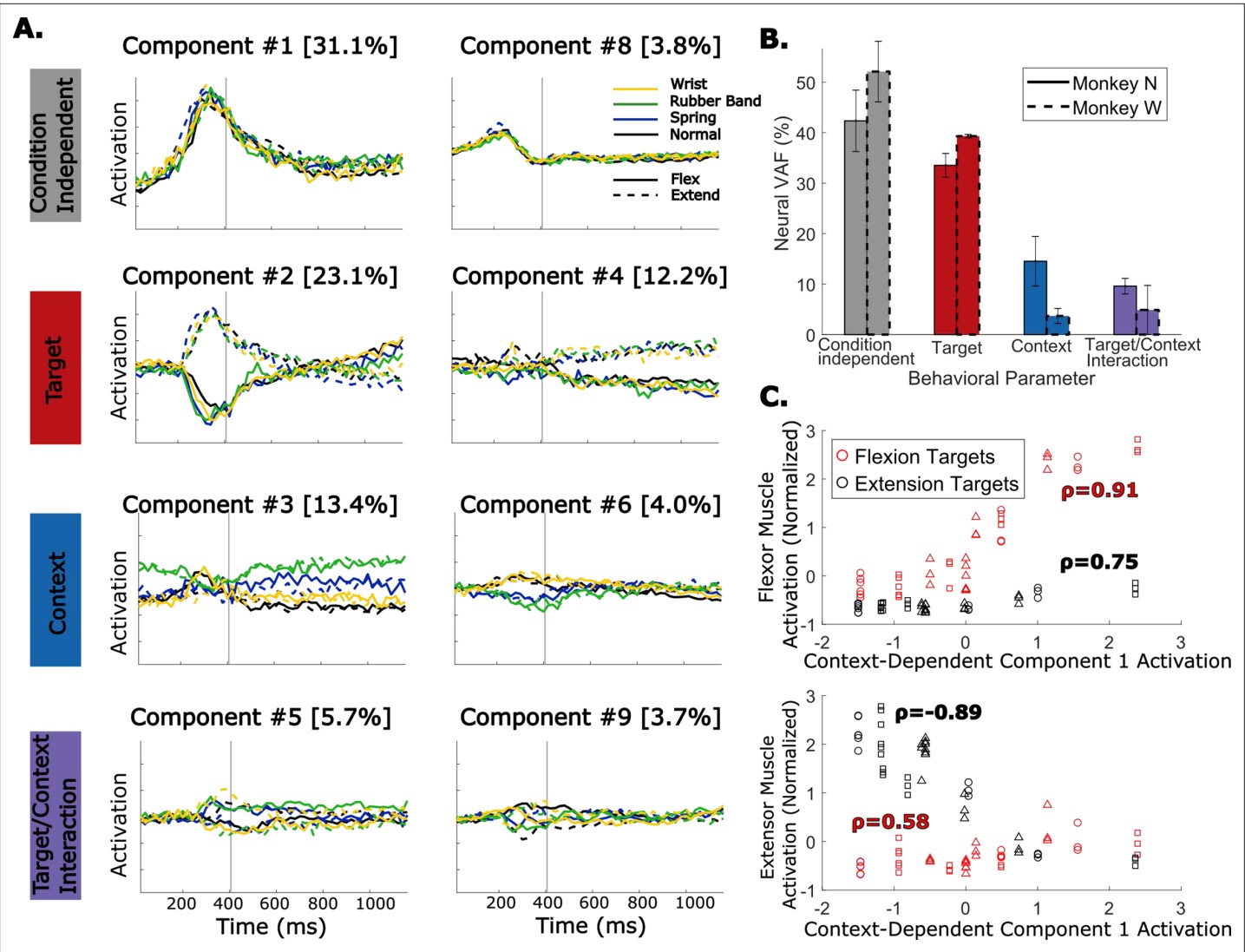

**Figure 8.** Dimensionally reduced representation of neural activity across multiple context changes. (**A**) Demixed PCA (dPCA) components for an example session with Monkey N performing normal trials (black), wrist trials (yellow), spring trials (blue), and rubber band trials (green). Solid traces are flexion trials, and dashed traces are extension trials. Components are organized by which behavioral parameter they explain the most neural variance for. Component numbers are ordered by how much neural variance they account for and the percent in brackets is the neural variance accounted for. (**B**) Percent of neural variance accounted for by each behavioral parameter. Bars with solid edges are for Monkey N, and bars with dashed edges are for Monkey W. Error bars indicate standard deviation across sessions. Three sessions are included for Monkey N, and two sessions are included for Monkey W. (**C**) Average muscle activation around peak movement for flexor muscles (top) or extensor muscles (bottom) during trials for each context and target plotted against the average activation of the first context-dependent dPCA component during the same trials. Each point represents the average activations in a series of trials in one context toward one target. Markers indicate which session the sample is from. Correlations are calculated within samples for one target, either flexion (red) or extension (black).

## Discussion

In this study we examined the impact of altering the context of a motor task, either adding an elastic resistance or postural change, while using a BMI for continuous finger control. These context changes represent a small sample of alterations found in activities of daily living but they include common changes to musculoskeletal properties of the hand during the task such as muscle tendon length and muscle activation range that give insight into how the results would extend to a wider range of changes. We found that changes in context increase the error of offline BMI decoder predictions significantly for both kinematics (*Figure 4*) and muscle activations (*Figure 3*). This effect was larger for predicting muscle activations than for predicting kinematics. In online trials using a kinematic-based

BMI, the monkeys were able to quickly adjust for context changes and achieved comparable performance to normal online trials. We tested this in two ways. First, we added context changes to the manipulandum during online trials (*Figure 5*), which resulted in almost no change in online performance. Second, we trained two decoders (one on normal trials and the other on off-context trials) and swapped between them for closed-loop control (*Figure 6*), which resulted in small but significant decreases in online performance for the model trained on off-context trials.

During the offline tasks, many channels changed neural activity with context, with 20.9–61.7% of tuned SBP channels modulating activity with context (*Table 1*). The magnitude of these shifts were relatively small, especially when compared to the large changes in required muscle activation (*Figure 2D–E*), with weak trends to require greater activation for resisted flexion and lesser for assisted extension (*Figure 7B–C*). Additionally, the neural manifolds underlying movements in each context were well aligned (*Figure 7D*). Using dPCA we found that while a large proportion of neural variance was explained by dPCA components that did not change with context, a significant proportion of the neural variance is associated with components that are context-dependent (*Figure 8B*). Visually, the context components are shifting the trajectories without changing the overall shape and the shift in neural activity is strongly correlated with muscle activations in new contexts (*Figure 8C*). This agrees with other studies which found lower variance activity may be related to the actual motor commands (*Gallego et al., 2018*; *Russo et al., 2018*; *Saxena et al., 2022*).

The similar online performance in each context, despite large offline mismatch, may be explained by a few possible factors. First would be if normal online trials are performed using a model that already does not capture the relationship between neural activity and intended finger movements well. In a control-systems perspective, online BMI control changes the 'plant' from native fingers to the virtual fingers on the screen. There is evidence that as a result of this, neural activity is different during online BMI control (*Carmena et al., 2003*; *Fan et al., 2014*; *Ganguly et al., 2011*; *Gilja et al., 2015*; *Jarosiewicz et al., 2013*; *Orsborn et al., 2012*; *Taylor et al., 2002*). Additionally, it is unlikely that a linear model like those used here robustly capture the relationship between motor cortex activity and kinematics. Due to the change in neural activity during online trials and inaccuracies in the decoding model, there is likely adjustment required by the monkey to use the BMI online during the normal setup. In this case, performing the online task in a new context is swapping one non-optimal decoder for a new one. In both BMI experiments, the initial BMI performance, that is the first five trials, was not worse for off-context trials as compared to normal online BMI trials, suggesting that both types of trials were of similar difficulty to adjust to (*Figures 5C and 6C*).

The similar online performance could also be observed if the context change does not have a large impact on task-relevant neural activity. Studies into neural plasticity have shown that during a session of online trials, subjects can adjust to decoder perturbations that are within the same intrinsic manifold (*Sadtler et al., 2014*). We found that individual channel activations change on up to 61.7% of channels that are important for decoding movements (*Table 1*), and this introduces error into model predictions. However, if the perturbations we introduced did not shift activity outside of the intrinsic manifold, then it may have been easy to adjust to the new context. Our data show a near instantaneous adaptation to the perturbations whereas Sadtler et al. found some within-manifold perturbations required on the order of hundreds of trials to adapt to, indicating that our perturbation was intuitive to adapt to. Analyzing the monkey's strategy during the BMI task revealed that they were able to do the BMI task with perturbations to the hand without adjusting their strategy (*Figure 5D*). This likely follows from the BMI task being driven by neural activity and visual feedback rather than movements of the hand itself. In the two decoder BMI task where off-context performance was often slightly worse, the monkeys did make small adjustments to perform the task (*Figure 6D*). For example, they tended to 'push' harder to flex the virtual hand during off-context trials. This re-aiming strategy is similar to what has been described in other work on short-term learning with motor BMI (*Golub et al., 2018*; *Jarosiewicz et al., 2008*). In this case re-aiming likely stems from the need to reproduce a higher or lower neural activation online in order to use a model trained on data where channel activation increased for flexion and decreased for extension for the spring context (*Figure 7B–C*).

The online BMI experiments in this study used a kinematic-based BMI decoder. BMI studies typically predict kinematic variables for applications such as prosthesis control (*Hochberg et al., 2012*; *Wodlinger et al., 2015*) and cursor or virtual movement control (*Gilja et al., 2015*; *Hochberg et al., 2006*; *Young et al., 2019*). In the offline predictions using linear models, we found that neither

kinematics nor muscle activations could be predicted at the same accuracy in new contexts. While significant, kinematics, specifically flexion velocity, did show a smaller decrease in offline performance between contexts (*Figure 4B–C*). These results suggest that when designing BMI, using kinematic variables as a command signal may allow for better generalization when the biomechanics of the task are not important, such as virtual tasks.

However, in FES applications (*Ajiboye et al., 2017*; *Bouton et al., 2016*; *Nason-Tomaszewski et al., 2022*), biomechanics are important. The final outputs are stimulation parameters that cause a desired amount of muscle contraction. Importantly, the required stimulation parameters could change with context due to the change in required muscle activation. As a result, even if predictions of position or velocity generalize well to new contexts, the mapping from kinematics to stimulation parameters would no longer be accurate. Our results with online BMI indicate that the monkeys are able to adapt by re-aiming with the BMI to restore some ability to do the virtual online task, which indicates they may also be able to re-aim in FES applications as well. However, in our task this adaptation occurred with a performance loss (*Figure 6B*). Instead it would be better to account for how context changes the biomechanics of the task with the BMI. This could be done either through incorporating a better control system into the BMI, for example developing a controller to update stimulation parameters to match the decoded joint angle or velocity, or by better estimating the intended muscle activations from neural activity. Decoded intended muscle activations can be mapped to stimulation parameters as done by some FES studies (*Ethier et al., 2012*; *Hasse et al., 2022*).

With regard to decoding intended muscle activations, non-prehensile finger movements are less studied than arm movements and grasping, partly due to experimental difficulty, with much work coming from only a few datasets (*Shah et al., 2009*; *Schieber, 1991*). Although predictions of muscle activations from neural activity for muscles overlapping with those in this study have been done for movements of the wrist (*Naufel et al., 2019*; *Oby et al., 2013*) and grasp (*Ethier et al., 2012*), predicting finger-related muscle activations during non-prehensile finger movements has to our knowledge not been attempted yet. Here, we found that we could decode muscle activations within each context during this individuated finger task with similar accuracy as decoding kinematics (*Supplementary files 1 and 2*).

Predicting muscle activation also led to the poorest offline generalization. The off-context predictions of muscle activation had both a large unaccounted for magnitude change and a lower correlation. We observed that neural features change by a relatively small magnitude (*Figure 7B–C*) whereas the muscle activation changes by large amounts (*Figure 2D–E*), resulting in linear models failing to predict a large enough scaling for off-context muscle activation. This observation matches studies of wrist movements where predicting muscle activation also did not generalize well (*Naufel et al., 2019*). The lower correlation was partially driven by muscle activation patterns not observed in normal context training data, such as increased co-contracting flexor and extensor muscles during flexion trials to modulate stiffness when springs were present, as seen in *Figure 3A* where the predicted EDC activation does not increase for flexion in the spring condition. With a better model, it might be possible to pick out the relationship between neural activity and muscle activations. Determining for example that the intention is to activate EDC more, whether that is to co-contract with FDP or to extend the fingers may not matter as long as the intention can be accurately decoded. Based on these results, it's likely that linear models are not able to pick out this relationship. Additionally, some prediction generalization error can be associated with the muscle activations being a higher dimensional variable than kinematics, with evidence that motor cortex can selectively activate or inactivate specific muscles (*Schieber et al., 2009*). In this study there was no effort to estimate lower dimensional muscle synergies that may be underlying the observed muscle activations, but it is possible that cortical activity would relate more linearly to muscle synergies than to individual muscle activations.

An alternative BMI design approach to decoding movements is to use task-specific features to augment decoder models (*Schroeder et al., 2022*). The context shifts studied here represent a small and discrete subset of the shifts found in activities of daily living, however they relate to continuous musculoskeletal properties that are shifting with the context, that is muscle length, co-contraction, or muscle activation magnitude. Identifying a feature in neural activity that accounts for the change in muscle activation across contexts would assist in decoder generalization. For example, the context-dependent neural activity that strongly correlated with muscle activations in new contexts (*Figure 8C*) could provide a feature for accounting for the scaling change while predicting muscle activation or

allow models to modulate force or muscle activation while producing the same kinematics. More work is needed to understand if a neural feature like this would remain stable in different cognitive contexts, for example grasping or freely moving fingers as opposed to doing this virtual target acquisition task.

Nonlinear models could also improve predictions of intended muscle activation or kinematics from neural activity. Complex models are becoming more widely used in order to better model the relationship between motor cortex and intended movements (*Glaser et al., 2020*; *Schwemmer et al., 2018*; *Sussillo et al., 2012*; *Willett et al., 2021*; *Willsey et al., 2022*). While a general concern for these nonlinear models is that they will overfit to the training data and not generalize well, given the correct training data they also may be able to identify less obvious trends that will distinguish between contexts and allow for better predictions. For example, *Naufel et al., 2019*, were able to predict muscle activations in multiple wrist tasks after training an LSTM decoder on data from all of the tasks. This indicates that there may be enough information in neural features to distinguish between the different tasks. Our dPCA results indicate that around 24% of neural variance can be accounted for by context-specific activity (*Figure 8B*), so it is likely that a neural network would be able to take advantage of that information to make predictions in multiple contexts. More work will be needed to characterize how much training data these nonlinear models need in order to generalize to all the contexts experienced during activities of daily living.

## Methods

All procedures were approved by the University of Michigan Institutional Animal Care and Use Committee (protocol numbers PRO00010076 and PRO00008138).

### Implants

We implanted two male rhesus macaques (Monkey N age 8–9, Monkey W age 8–9) with Utah micro-electrode arrays (Blackrock Microsystems, Salt Lake City, UT, USA) in the hand area of precentral gyrus, as described previously (*Irwin et al., 2017*; *Nason et al., 2021*; *Vaskov et al., 2018*). Two monkeys were chosen to ensure results are consistent between subjects. Monkey N was implanted with two 64-channel arrays in right hemisphere primary motor cortex and Monkey W was also implanted with two 96-channel arrays in right hemisphere primary motor cortex. Channels from both of Monkey N's motor cortex arrays and from Monkey W's lateral motor cortex array were used in this study, for a total of 96 channels from each monkey for analysis. The number of channels simultaneously recorded was limited to 96 due to the available recording hardware. Monkey N was between 511 and 1168 days post-cortical implant and Monkey W was between 254 and 411 days post-cortical implant during data collection.

Monkey N was also implanted with chronic bipolar intramuscular EMG recording electrodes (Synapse Biomedical, Inc, Oberlin, OH, USA) in a separate surgery as described previously (*Nason et al., 2021*). The list of muscles targeted along with their function are included in *Table 2*. Briefly, muscles were accessed via dorsal and ventral incisions on the left forearm and specific muscles were surgically identified with the assistance of intraoperative stimulation. Bipolar electrodes were inserted and sutured into the muscle belly near the point of innervation and then tunneled over the elbow and shoulder to an interscapular exit site. The percutaneous electrodes were connected to a 16-channel

**Table 2.** List of muscles targeted during surgery with their associated function.

| Muscle | Function |
| --- | --- |
| Extensor indicis proprius (EIP) | Index finger extensor |
| Flexor digitorum profundus, targeting MRS (FDP) | Finger flexor |
| Extensor digitorum communis (EDC) | Finger extensor |
| Extensor carpi radialis brevis (ECRB) | Wrist extensor |
| Flexor carpi ulnaris (FCU) | Wrist flexor/adductor |
| Flexor carpi radialis (FCR) | Wrist flexor |
| Flexor digitorum profundus, targeting index, proximal and distal sites (FDPip, FDPid) | Index finger flexor |

PermaLoc connector. Monkey N was between 120 and 496 days post-EMG electrode implant for all EMG data collected.

## Feature extraction

(TCFR and SBP) were recorded in real time during experiments using the Cerebus neural signal processor (Blackrock Microsystems). Threshold crossings were acquired by configuring the Cerebus to threshold each channel at –4.5 times the signal root-mean-square. For each threshold crossing, spike snippets were sent to a computer running xPC Target version 2012b (Mathworks) which saved the channel and time for each threshold crossing. SBP is an estimate of power in the 300–1000 Hz frequency band and was acquired by configuring the Cerebus to bandpass filter the raw signals to 300–1000 Hz using the Digital Filter Editor feature in the Central Software Suite version 6.5.4 (Blackrock Microsystems), then sample at 2 kHz. The filtered 2 kHz recording was then sent to the computer running xPC Target, which rectified and summed the samples on each channel received in each 1 ms iteration and counted the quantity of samples received each 1 ms so that SBP could later be averaged within longer time bins. Both the threshold crossings and SBP were saved by xPC synchronized with other real-time experimental information. Artifacts were removed for TCFR by removing threshold crossing times if 20 or more channels had threshold crossings in the same millisecond. Features were binned into non-overlapping bins of length 32 ms for online and offline decoding, or bins with a length of 20 ms for calculating tuning and comparing features across trials. SBP is summed for every 1 ms in the time bin and then divided by the total number of raw 2 kHz samples in the bin. For TCFR the spike counts are summed within a bin and then divided by the bin size to get a threshold crossing rate.

EMG from Monkey N's eight bipolar electrodes was recorded for later offline synchronization. The percutaneous PermaLoc connector was connected to a CerePlex Direct (CPD) via a 64-channel splitter box and CerePlex A (Blackrock Microsystems) which converted the signals to the digital domain with unity gain. The CPD was configured to record 16 channels of raw signal at 10 kHz and for each bipolar pair the electrode implanted further inside the muscle was software referenced to the second electrode. These eight bipolar referenced channels are used in analyses. To synchronize EMG offline, we used the Sync Pulse functionality in Central to create unique pulses that were recorded by both the Cerebus and CPD and could later be used to align the Cerebus and CPD recordings. For offline analysis, muscle activations are estimated from the 10 kHz EMG recording by filtering with a second-order Butterworth bandpass filter between 100 and 500 Hz and then taking the mean absolute value of the filtered signal during every binning period.

## Experimental setup

During experiments the monkeys performed a virtual finger task while motor cortex activity and optionally arm muscle activity were recorded as described. Similar to previously described experiments (*Irwin et al., 2017*; *Nason et al., 2021*; *Vaskov et al., 2018*), we used xPC Target to coordinate the experiment in real time. The xPC Target computer acquired and stored task parameters and neural features in real time, coordinated target presentation, acquired finger positions from the flex sensors on each finger group (FS-L-0073-103-ST, Spectra Symbol, Salt Lake City, UT, USA), and sent finger positions and target locations to a computer simulating movements of a virtual monkey hand (MusculoSkeletal Modeling Software) (*Davoodi et al., 2007*). For online experiments, the xPC Target computer also binned threshold crossings and SBP in customizable bin sizes and evaluated the decoder model to predict finger positions in real time using an RFKF (see details below).

## Behavioral task

Monkeys N and W were trained to acquire virtual targets by moving their physical fingers in a manipulandum to control virtual fingers on a screen in front of them. All sessions took place in a shielded chamber with the monkey's head fixed and arms restrained at their side with elbows bent 90 degrees and hands resting on a table in front of them. The left hand was placed in a manipulandum described previously (*Nason et al., 2021*), with openings separating the index finger and the MRS finger group (*Figure 1C*). The monkeys were trained to move the index finger independently of the MRS finger group (*Figure 1B*), that is 2-DOF, although in some trials they moved both finger groups as 1-DOF. Each trial began with spherical targets appearing for each active finger group with each target

occupying 15% of the full range of motion of the fingers. In the 1-DOF task the target was presented to the index finger.

Target presentation followed a center-out-and-back pattern with every other target presented at a center position, equivalent to 50% on a scale from 0% (full extension) to 100% (full flexion). Additionally, center was presented after any failed trial. The non-center targets were randomly selected from a set of targets. For 2-DOF the targets included any combination of index flexion, rest, or extension, and MRS flexion, rest, or extension, with a randomly chosen magnitude of 20%, 30%, or 40% of the full movement range. The split movements (index flexion with MRS extension or vice versa) did not have a 40% movement magnitude because the monkeys had difficulty splitting the finger groups that far. The 1-DOF movements were also center-out with the fingers flexing or extending either 40% from rest or a randomly chosen magnitude of 20%, 30%, or 40% from rest depending on the session, the former generally being used for tuning analyses and offline comparisons and the latter being used for online experiments. In each trial the monkey had to hold their fingers within the target(s) for 750 ms. During online decoding experiments, the same center-out-and-back target presentation order was used but the hold time was reduced to 500 ms. In one session used in offline analysis and four online sessions, the hold time was 2 ms longer than expected due to a minor bug. In two online experiments with the manipulandum context changing BMI task, targets were presented in a random order instead of center-out. In this target presentation, a target separation up to 50% of the movement range and a center position were randomly generated and one target for each finger group was presented at the generated target separation from each other, equidistant from the generated center position.

Task context was altered through four potential task alterations. One alteration was the addition of torsional springs to both finger groups (180 degree deflection angle, 0.028 in or 0.04 in wire diameter, Gardner Spring Inc, Tulsa, OK, USA), referred to as the 'spring context'. The second alteration was the rotation of the manipulandum by 23 degrees in the flexion direction, referred to as the 'wrist context'. A third alteration was introduced by attaching rubber bands from the back of the manipulandum to the door for each finger group, thereby resisting flexion, referred to as the 'rubber band context'. A last alteration was addition of torsional springs and the rotation of the manipulandum by 23 degrees at the same time, referred to as the 'both context'. Trials performed with one of these alterations are referred to as 'off-context' trials, while trials performed without alterations are referred to as 'normal' trials. As the index finger alone is much weaker than the MRS finger group, the index finger used a smaller spring when applicable. The added springs increased the force required for full flexion by 9.5 N (for MRS) and 3.3 N (for index), while the rubber bands increased the force required for full flexion by 16.5 N. The rubber band context was only done by Monkey N and in a 1-DOF task due to task difficulty. For reference, full flexion required approximately 1.3 N of force without the springs or bands.

## Comparison of kinematics and muscle activation between contexts

Three representative sessions for both Monkey N and Monkey W, 1 day for each context – spring or wrist – were used to compare kinematics across contexts. During data collection, normal trials and off-context trials were interleaved by alternating context type every 175–350 trials in order to control for changes in behavior over time. During these representative days, there was an average of 1134 normal trials and 1118 off-context trials per day for Monkey N and 526 normal trials and 504 off-context trials per day for Monkey W. To compute how finger velocity changed between normal trials and off-context trials, the peak velocity of finger movements was found for every trial. For every trial, the recorded finger flexions were downsampled to 20 ms and filtered with a second-order Savitzky-Golay FIR filter. Finger velocity was estimated from the downsampled and filtered finger positions and maximum finger speeds were found. The peak movement time was taken at the time of the largest peak in speed after trial start. Trials were then split by context and target direction (flexion vs. extension), and a two-sample t-test was used to compare peak speeds and compute a 99% confidence interval, once for flexion targets and again for extension targets. Comparisons were made only between trials to the same target, leaving about 281 trials per group and 129 trials per group for each comparison for Monkey N and Monkey W, respectively.

The same sessions for Monkey N used to compare kinematics were also used to compare muscle activations. The recorded EMG was filtered and the mean absolute value was taken in 20 ms bins as described previously. Binned muscle activations were then smoothed with a 100 ms Gaussian kernel.

One value was obtained for every trial by taking the average muscle activation in a 420 ms window around peak movement, including 10 bins before peak movement, the bin that included peak movement, and 10 bins after peak movement. These muscle activation values were grouped by context and target, then compared with a two-sample t-test.

## Computation of neural tuning and context modulation

During five representative experiments for each monkey, three that tested the spring context and two that tested the wrist context, we calculated the number of channels that were significantly modulated by any finger movement and the number of channels with a change in activity between normal trials and off-context trials. During these sessions, the monkeys performed the task with all fingers moving together (1-DOF), in a center-out task as described, to targets at either plus or minus 40% from center. Trials that were unsuccessful and trials following unsuccessful trials were removed. Unsuccessful trials were rare, often only occurring on the first or last trial of a block of trials. There was an average of 1072 normal trials and 747 off-context trials for Monkey N and 544 normal trials and 387 off-context trials for Monkey W were used during these sessions.

Channel tuning and context modulation was calculated with both the SBP features and TCFR features. On each day, features and kinematics were averaged into non-overlapping 20 ms bins, data from normal trials and off-context trials were concatenated together, and the SBP and TCFR were each normalized to zero mean and unit standard deviation. An optimal lag was calculated for each channel by maximizing the L2-norm of regression coefficients between a feature and finger position and velocity. Features at that optimal lag were then regressed with finger position and velocity one at a time with an added effect for context following these equations:

$$X_n = \begin{bmatrix} \hat{x}_n & c\hat{x}_n \end{bmatrix}$$
$$Y = B + X_n W_n$$

where $\hat{x}_n$ is the T × 1 vector containing T bins of channel SBP or TCFR for channel n, c is an indicator variable that equals one if that sample was during an off-context trial or zero otherwise, Y is a T × 2 matrix containing finger position and velocity, B is the trained linear offset, and $W_n$ is the 2×2 matrix of trained weights relating channel n's activity to finger position and velocity. A channel was called tuned if the regression coefficient between the neural feature and either finger position or velocity, that is $w_{1,1}$ or $w_{1,2}$, were significantly different from zero, via a t-test on the regression coefficient. A channel was also called context modulated if either coefficient in the second row of $W_n$, which includes the effect of context, was significantly different from zero, also via a t-test, indicating a different slope relating neural activity and kinematics between normal trials and off-context trials. False discovery rate correction was applied to each session at a level of 0.1%.

To quantify the change in neural activity between contexts as in *Figure 7*, we used one representative session for Monkey N in which trials were done in the normal, spring, wrist, and rubber band contexts. An additional two representative sessions for Monkey W were used, one session comparing normal and spring trials and another session comparing normal and wrist trials, both performed with 1-DOF movements and with targets to 40% flexion or extension from rest only. Tuned channels were calculated as previously described using the SBP feature. For every trial, the SBP was binned into 20 ms bins and then smoothed with a 100 ms Gaussian kernel. Then the average activity in a window spanning 200 ms before and 200 ms after the bin containing peak movement was calculated for each tuned channel. The trials were then split by context and by target, and the trial SBP values were compared between contexts with a two-sample t-test.

## Offline predictions

Data from nine sessions with Monkey N, three for each context (springs, wrist, and both), were used for offline muscle activation and kinematic predictions, and three sessions for Monkey W, one for each context, were used for offline kinematic predictions. During these sessions, both monkeys performed the 2-DOF center-out task. Blocks of normal trials and off-context trials were interleaved by alternating context in order to control for changes in neural activity over time. Trials that were unsuccessful were removed before analysis. There was an average of 803 normal trials and 470 off-context trials for Monkey N and 737 normal trials and 329 off-context trials for Monkey W. To account for changes in monkey motivation, sessions chosen were those with consistent prediction accuracy between early

and late normal trials within a session. These sessions spanned 165 days starting 792 days post-cortical array implant and 120 days post-EMG electrode implant for Monkey N, and 63 days starting 285 days post-cortical array implant for Monkey W.

In each session, SBP and muscle activations or kinematics were binned into 32 ms bins and features were concatenated across trials of the same context. The SBP channels were masked to those with an average TCFR greater than 1 Hz across a session and 12 bins of history from each of these channels were used as additional features. Ridge regression relating SBP to muscle activations or kinematics was trained on normal trials and then tested on both normal trials and off-context trials with 10-fold cross-validation. To do this, the normal trials were split into 10 folds with an equivalent number of bins in each fold, a model was trained on nine folds, and then tested on the left-out fold as well as on data from off-context trials. We used two metrics to evaluate prediction accuracy. First we used the Pearson correlation coefficient between the predicted and measured muscle activations or kinematics to establish how well the predictions are linearly correlated with measurements. The second metric was MSE normalized by the variance of the measured data (MSE). Normalizing by the variance allows for better comparison across test datasets as they may have different variances. In this formulation, the MSE is the fraction of unexplained variance or one minus the variance accounted for or coefficient of determination used in previous studies (*Fagg et al., 2009*; *Naufel et al., 2019*). Values greater than one indicate that the predictions are introducing variance compared to the worst possible least-squares predictor, that is predicting the mean.

## Online decoding

We used either a KF or an RFKF (*Gilja et al., 2012*) to predict intended finger movements for all BMI experiments, as done previously (*Irwin et al., 2017*; *Nason et al., 2021*; *Vaskov et al., 2018*). We performed two types of online experiments. In the first experiments, an RFKF was trained on normal trials and then used during trials with context changes in the manipulandum or without any additions to the manipulandum. To train the model, monkeys first performed at least 300 trials of center-out manipulandum control with 750 ms hold time. Using these trials, we trained a position/velocity KF which the monkeys used online for at least 200 trials, with a 32 ms update rate and a 500 ms hold time. To use the KF, virtual finger position was updated by integrating the predicted velocity in the current time step to update the previous step's finger position. An RFKF was then trained, as done previously (*Nason et al., 2021*), by rotating incorrect velocities during online control with the KF to be toward the intended target represented in a two-dimensional space, setting finger velocity equal to zero when in the correct target, and then retraining regression coefficient matrices. The RFKF was used online for blocks of 100–200 trials with different context changes applied to the manipulandum, alternating between normal trials and other contexts. Multiple contexts could be tested in one session during these experiments by switching out the context manipulations present in the manipulandum.

In the second set of online experiments, two KFs were trained in one session and then used alternatingly in online control without any changes present in the manipulandum. During these sessions, the monkeys first performed at least 300 trials of center-out manipulandum control, followed by another 300 or more trials of center-out manipulandum control with a context change present. One model was trained using each set of trials. The monkeys then used these models in online control for sets of 100–200 trials, and then the models were alternated. Hold times and update rates were kept consistent between types of experiments and sessions.

## Online performance measures

We estimated online performance with acquisition time, time to target, orbiting time, and path efficiency. Acquisition time was measured as the total time from target presentation to the end of the trial minus the hold time, therefore ending with the target being successfully acquired. Time to target was taken as the time from target presentation to the first time where all fingers with targets were in their targets. Orbiting time was then calculated as the time from all fingers first reaching their targets to the end of the trial minus the hold time. Trials where the fingers reached the targets and never left therefore had an orbiting time of 0 ms. Failed trials were excluded when comparing online performance between context but not for evaluating the monkeys adaptation within the first five trials. Path efficiency was calculated as the ratio of the shortest distance between the fingers' starting positions

and the target positions projected onto a two-dimensional space, to the length of the path traveled by the fingers.

## Online neural activity patterns

To visualize neural activity during online trials, the normal offline training trials used to train the KF were used to train a new linear readout between neural activity and finger velocities. Neural activity from these trials was binned in 50 ms intervals, then neural activity in the current bin and the five most recent bins were regressed with the finger velocities during these trials to obtain one set of weights for that session. This model was then used to predict velocities from neural activity during the online trials. The predictions at the time point in the trial with the peak online velocity toward the target during the online trial were taken as the online neural activity patterns. To compare neural activity patterns across multiple targets in multiple sessions, the neural activity patterns for each trial were projected onto the target direction for each trial to obtain one 'pushing magnitude', or the velocity magnitude that they were pushing toward the target direction. Pushing magnitudes were collected for each trial, separated for flexion trials (IF, MF, IF+MF), extension trials (IE, ME, IE+ME), and split trials (IF+ME, IE+MF), and then the pushing magnitudes for each set of trials were compared between normal trials and off-context trials using a two-sample t-test.

## Dimensionality reduction

To investigate changes in population neural activity due to changes in context, two sessions of 1-DOF center-out trials with targets of 40% flexion or extension from rest were used for each monkey. For Monkey N, both sessions included trials where the task was performed in the normal, spring, rubber band, and wrist contexts. For Monkey W, one session included trials in the normal and spring contexts, and the other session included trials in the normal and wrist contexts. SBP was binned into 20 ms bins, masked to only include channels with TCFR greater than 1 Hz, and then for each trial a time frame 400 ms before to 740 ms after the bin containing peak movement was taken from each trial. The neural activity for trials within a single context was concatenated and averaged across trials with the same context and target forming an $N \times T \times D$ data structure for each context, where N is the number of channels, T is the number of bins per trial used, and D is the number of targets. Neural data was then concatenated across targets to form an $N \times (T*D)$ matrix and then we used PCA to calculate a manifold for each context, keeping the top 16 components for Monkey N and eight components for Monkey W, which explained 86% of variance on average. Principal angles were found between the manifolds following methods used previously (*Björck and Golub, 1973*; *Gallego et al., 2018*). These principal angles are the minimal angles required to align the manifolds and serve as a measure for how well aligned two manifolds are. As a control, two sets of 50 trials were taken from the normal trials and used to calculate two manifolds in the same way. The principal angles between these manifolds were then calculated. The sampling and angle calculations were repeated 100 times to obtain a control distribution of principal angles.

We also calculated one manifold spanning trials from all contexts tested in one session. This was done using dPCA (*Kobak et al., 2016*). This approach finds a single neural manifold that reduces the dimensionality of the data while maintaining a linear readout that can reconstruct the mean neural activation associated with manually chosen behavioral variables. In this instance, the behavioral parameters chosen were target, that is either flexion or extension, and which context the task was done in. MATLAB code for calculating dPCA components was downloaded from http://github.com/machenslab/dPCA, SBP was binned into 20 ms bins, masked to include only channels with TCFR greater than 1 Hz, and then concatenated into an $N \times C \times D \times T \times n$ data structure where N, D, and T follow the same structure as the PCA calculations, n is the number of trials per condition, and C is the number of contexts tested in that session. SBP was averaged over the number of trials, n, to form the peristimulus-time-histograms for each target and context combination, after which dPCA components were calculated. Neural variance of a behavioral parameter was obtained by calculating the variance within the marginalization of neural data based on each behavioral parameter and taking the ratio of the total variance in a marginalization to the total variance in the neural data.

## Acknowledgements

We thank Eric Kennedy for animal and experimental support and the University of Michigan Unit for Laboratory Animal Medicine for expert veterinary and surgical support. We appreciate Chris Andrew's expert statistical assistance and the support of the University of Michigan Biointerfaces Institute.

## Additional information

### Funding

| Funder | Grant reference number | Author |
|---|---|---|
| National Science Foundation | Grant Number 1926576 | Matthew J Mender Hisham Temmar Parag Patil Cynthia A Chestek |
| National Science Foundation | Graduate Research Fellowship Program | Joseph T Costello |
| Eunice Kennedy Shriver National Institute of Child Health and Human Development | Grant Number F31HD098804 | Samuel R Nason-Tomaszewski |
| National Institute of Neurological Disorders and Stroke | Grant Number T32NS007222 | Matthew S Willsey |
| National Institute of Neurological Disorders and Stroke | Grant Number R01NS105132 | Nishant Ganesh Kumar Theodore A Kung |
| The D. Dan and Betty Kahn Foundation | Grant AWD011321 | Dylan M Wallace |
| University of Michigan Robotics Institute | Graduate student fellowship | Dylan M Wallace |
| A. Alfred Taubman Medical Research Institute | | Parag Patil |
| Craig H. Neilsen Foundation | Project 315108 | Cynthia A Chestek |
| National Institute of General Medical Sciences | Grant Number R01GM111293 | Parag Patil Cynthia A Chestek |

The funders had no role in study design, data collection and interpretation, or the decision to submit the work for publication.

### Author contributions

Matthew J Mender, Conceptualization, Software, Formal analysis, Validation, Investigation, Methodology, Writing - original draft, Writing - review and editing; Samuel R Nason-Tomaszewski, Conceptualization, Software, Investigation, Methodology, Writing - review and editing; Hisham Temmar, Joseph T Costello, Dylan M Wallace, Validation, Investigation, Writing - review and editing; Matthew S Willsey, Nishant Ganesh Kumar, Theodore A Kung, Writing - review and editing; Parag Patil, Conceptualization, Supervision, Funding acquisition, Writing - review and editing; Cynthia A Chestek, Conceptualization, Supervision, Funding acquisition, Methodology, Writing - review and editing

### Author ORCIDs

Matthew J Mender (ID) http://orcid.org/0000-0003-1562-3289
Hisham Temmar (ID) http://orcid.org/0000-0002-4464-4911
Cynthia A Chestek (ID) http://orcid.org/0000-0002-9671-7051

## Ethics

All protocols were in accord with the National Institutes of Health guidelines and approved by the University of Michigan Institutional Animal Care and Use Committee (protocol numbers PRO00010076 and PRO00008138).

## Decision letter and Author response

Decision letter https://doi.org/10.7554/eLife.82598.sa1
Author response https://doi.org/10.7554/eLife.82598.sa2

---

## Additional files

### Supplementary files

• Supplementary file 1. Average prediction correlation when using ridge regression models to predict muscle activations or kinematics with the same or different test contexts. Results for Monkey N using the same sessions included in *Figure 3* and *Figure 4* and for Monkey W using the same sessions included in *Figure 4*.

• Supplementary file 2. Average prediction mean-squared error (MSE) when using ridge regression models to predict muscle activations or kinematics with the same or different test contexts. Results for Monkey N using the same sessions included in *Figure 3* and *Figure 4* and for Monkey W using the same sessions included in *Figure 4*.

• MDAR checklist

### Data availability

Neural, behavioral, EMG, and online BMI performance data has been deposited in the Dryad repository (https://doi.org/10.5061/dryad.p2ngf1vtn).

The following dataset was generated:

| Author(s) | Year | Dataset title | Dataset URL | Database and Identifier |
|---|---|---|---|---|
| Mender MJ, Nason-Tomaszewski SR, Temmar H, Costello JT, Wallace DM, Willsey MS, Ganesh Kumar N, Kung TA, Patil P, Chestek CA | 2023 | Data from: The impact of task context on predicting finger movements in a brain-machine interface | https://doi.org/10.5061/dryad.p2ngf1vtn | Dryad Digital Repository, 10.5061/dryad.p2ngf1vtn |

---

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
