## [Editor Report]

This study provides valuable findings about how brain machine interfaces cope with changes in context, an important consideration for deploying such devices in the real world. The evidence supporting the claims is solid, and the findings will be of interest to motor neuroscientists and engineers developing brain machine interfaces.

---

## [Decision Letter]

**Decision letter after peer review:**

Thank you for submitting your article "The Impact of Task Context on Predicting Finger Movements in a Brain-Machine Interface" for consideration by *eLife*. Your article has been reviewed by 3 peer reviewers, and the evaluation has been overseen by Andrew Pruszynski as the Reviewing Editor and Tamar Makin as the Senior Editor. The following individual involved in review of your submission has agreed to reveal their identity: Christian Éthier (Reviewer #1). We apologize for the delay in returning these reviews.

Essential revisions:

1. Provide a convincing explanation as to why the two BMI experiments (Figure 5A and B) gave different results. This is an important consideration as it forms the basis for a key conclusion of the study. This likely requires further data analysis.

2. The authors need to dive deeper into their data, specifically that which surrounds contextual shifts and the associated adjustment. There is a lot left on the table here that could broaden the scope and impact of the present study. All reviewers raise this point. For example, Reviewer 3 asks whether this adjustment reflects a 're-aiming' strategy. Again, this likely requires further data analysis.

3. The authors should provide a clearer description of how the present findings, which are in a very limited and narrow set of conditions, further BMI design principles in general. That is, how do the authors think the present results acquired in a limited context will generalize to the much wider set of contextual shifts that would be encountered in the real world.

4. There are a number of typos and general quality concerns with respect to the figures and the associated captions that are at the level that negatively influence the interpretability of the study and its findings. Specific suggestions are provided by the individual reviewers but a thorough revision is required in this respect.

*Reviewer #1 (Recommendations for the authors):*

A figure with pictures or drawings of the manipulandum in different context would be useful to better understand the experiment. Same for the virtual task/virtual hand. I'm not sure what the monkey

Perhaps the main body of the manuscript should contain a few words about what it means for channels to be 'tuned'. What criterion? Same for the definition of "changing activity".

On line 205 and Figure 2: if the spring resist flexion, why does the monkey activate is extensor muscle during flexion? To resist flexing too much? In this condition, is there tonic extensor muscle activity at the beginning of the trials? Moreover, this change in extensor and flexor synergy means that the monkeys not only scale their muscle activity for a different load, but seem to change their motor strategy. The different contexts thus require a non-linear change in muscle activity, but require more or less the same kinematics. How this impacts the decoders ability to generalize should be discussed.

Line 252-253: "This overall lack of change was surprising…" I fail to see how changing the hand configuration should impact online BMI performance. Because of differences in afferent, proprioceptive activity reaching the motor cortex? If the task is purely virtual, and the decoder good, the monkey can simply continue to try doing the "normal" task, rely on virtual feedback and should succeed. The "plant" in this case, does not change. It would be good to clarify if and how this was really an unexpected result.

Line 312: it would be helpful if the "window" was described here within the main manuscript

Figure 7: make it clear in the axes labels of panel C that "component 1 activation" refers to the first context-dependent component, not the first PC.

Methods line 561: indicate that the rotation is the direction of wrist flexion.

*Reviewer #2 (Recommendations for the authors):*

1. The authors state that BMI performance was minimally affected by context shifts because the animals were able to make fast adjustments online. However, they do not really dig into this adjustment. I recognize that there are only a few adjustment trials per context shift, but given the number of days and shifts on some of those days, I would imagine one could still examine the process of this adaptation in some detail. There are some nice examples of this in the literature, for example Golub et al. 2018 "Learning by neural reassociation" and Athalye et al. 2018. "Emergence of Coordinated Neural Dynamics Underlies Neuroprosthetic Learning and Skillful Control".

This seems like a missed opportunity to shed light on what cognitive strategies and/or neural tuning shifts. That in turn would lead to a more satisfying deeper story, which right now is essentially "neural activity changes between contexts, but BMI performance is okay, whew!".

2. As I alluded to in my summary statement, the difference in results between Figure 5A and 5B are quite substantial; the decoder trained off-context was quite a bit worse, and my view is that the current writing does not bring this finding to sufficient attention (e.g., the next paragraph starts with "To help explain how the monkeys were able to adjust to different contexts so well…"). A convincing explanation of why the two BMI experiments gave different results was not provided, but would be helpful for establishing confidence in the key conclusion. Rather, there's a greater focus on the lack of detrimental effect in figure 5A. To further explain my perspective: if this was a decoder innovation paper, a 32.6% improvement in times to target would be a big deal! So the performance decrease observed (and the asymmetry in what decoder+context mismatch is or is not compensated for) seems noteworthy. The Discussion should address this result, compared to the current "In both cases, the monkeys adjusted for the new context as quickly as they adjusted to normal online BMI trials."

3. Figure 1C – can the authors comment on why they think there are rather different changes in peak velocity between the two monkeys in response to the same experimental manipulations (e.g., opposite sign effect for flexion in the wrist condition)?

4. Lines 259-260 "This indicates that during the off-context online trials they adjusted and moved their hand differently to account for the context change". Couldn't it also be that they tried to move their hand the same as in the normal context (i.e., send the same descending motor command, with correspondingly the same firing rate patterns that the decoder picks up on and decodes "normally"), but the mechanical perturbation (e.g., springs) resulted in different hand movements. Said another way: imagine the (hypothetical, thought-experiment) scenario where there's no sensory feedback and the monkey is doing this all open-loop using whatever strategy they learned for the normal condition BCI. How are your Figure 4D results different from what we'd expect in this scenario?

5. Line 371-372: "In both cases, the monkeys adjusted for the new context as quickly as they adjusted to normal online BMI trials". Where was this shown? This sounds like a quantitative statement about how quickly performance reached the same level; but in the two-model (Figure 5b) tests off-context performance was substantially worse. And, I don't think the rate of adaption to normal BCI blocks was reported anywhere? Perhaps the authors meant something else and can rephrase to clarify?

6. Results are mostly consistent across both monkeys, but not every experiment or analysis is available from both muscles (e.g., EMG was only recorded from monkey N). While all the necessary information is present in the manuscript (e.g., via tables, figure legends, methods), I think the manuscript could be more clear in the main text to clarify which results are one-monkey results and which are two-monkey results. This helps the reader better assess the strength of evidence for each finding.

*Reviewer #3 (Recommendations for the authors):*

I have a number of specific.

1) It was not very easy to visualize what exactly the manipulandum was or what the effects of the springs were. A methods figure, perhaps as supplemental information would be useful. Related to this, it was not clear what the actual biomechanical effect of the springs were. Did they resist motion at specific joints, or across the entire digit. Can you also provide some details on the relative scale of the change in force (9.5N) to the range of grasp forces that normally be applied by the animal, or that would typically be applied in this task?

2) Based on the comment in the public review about whether M1 represents or generates movements, I would make two potential suggestions. You could simplify/reduce this text and streamline the introduction, or you could weave this idea and narrative throughout the manuscript, returning to it in the discussion. As it stands, this potentially important idea is raised, but then largely ignored for the rest of the paper.

3) In Figure 1D, what is the explanation for significantly increased activity in finger extensors during flexion in the spring context? Is this evidence of increased digit stiffness? If so, this could have interesting effects on population activity, as there is some evidence that stiffness control might be encoded in the cortex.

4) The method of calculating the Acquisition Time was not clear. Specifically, please provide a clearer description of how each dot in figure 4C and 5C are calculated. In the text description of 4C and 5C, it was difficult to interpret the text regarding which differences were significant and which were not, and what the differences in the text were referring to. For example, in Lines 263-266, it states that Monkey N had a short adaptation time (p = 0.002) and that Monkey W did not (p=0.22). I can't tell what exactly this is referring to in Figure 4C. Lastly, around line 290-292, it is stated that monkey W had no significant adaptation period. While I understand that the median z-score was close to 0, the range of z-scores in all cases in Figure 4C and 5C are very large. Were these randomly distributed across time? Was there a pattern? Can you say anything at all about the range of the z-scores themselves?

5) In the text, Figures 4D and 5D are discussed before 4C and 5C. Consider reorganizing the text or figure layout.

6) Regarding Figures 4D and 5D, is it possible to examine the distributions of the correlations in the normal and off-context trials and show this as error bars on each point? Can you statistically test whether the correlations really were different from each other to help support the claims in lines 280-290?

7) Throughout the discussion, it would be helpful to refer back to specific figures to support claims about the results of the manuscript.

8) The description of SBP data processing is unclear around Lines 497 and 504. What does it mean to sum the samples and track the quantity of samples? Please elaborate on what was done.

9) EMG channel differencing is unclear. On Lines 512-514, it sounds like you just did the difference of the 2 channels, which is fine. Was this done digitally after the fact? Please explain. This may help clarify the statement about also recording the 'partner' electrode.

10) Line 517: Provide additional details on the bandpass filter. What was the type and order of the filter?

11) Line 645: How common were unsuccessful trials?

---

## [Author Response]

Essential revisions:1. Provide a convincing explanation as to why the two BMI experiments (Figure 5A and B) gave different results. This is an important consideration as it forms the basis for a key conclusion of the study. This likely requires further data analysis.

This is an excellent point, we added additional explanatory text and analysis to the results to help explain why the BMI task with context changes added to the manipulandum did not see a large impact. As described in Common Response #2, we have added analysis related to how the monkeys adapt to the online task. Additionally, we bring up the difference between the two online experiments in the discussion now, starting line 486:

“Our data show a near instantaneous adaptation to the perturbations whereas Sadtler et al. found some within-manifold perturbations required on the order of hundreds of trials to adapt to, indicating that our perturbation was intuitive to adapt to. Analyzing the monkey’s strategy during the BMI task revealed that they were able to do the BMI task with perturbations to the hand without adjusting their strategy (Figure 5D). This likely follows from the BMI task being driven by neural activity and visual feedback rather than movements of the hand itself. In the two decoder BMI task where off-context performance was often slightly worse, the monkeys did make small adjustments to perform the task (Figure 6D). For example, they tended to “push” harder to flex the virtual hand during off-context trials.…”

2. The authors need to dive deeper into their data, specifically that which surrounds contextual shifts and the associated adjustment. There is a lot left on the table here that could broaden the scope and impact of the present study. All reviewers raise this point. For example, Reviewer 3 asks whether this adjustment reflects a 're-aiming' strategy. Again, this likely requires further data analysis.

This is an excellent suggestion. Using the analysis of online adaptation described in Common Response #2, we found that there are small changes to the monkey’s strategy using the BMI. We have reframed a discussion paragraph to include this adaptation analysis, starting line 480:

“The similar online performance could also be observed if the context change does not have a large impact on task-relevant neural activity. Studies into neural plasticity have shown that during a session of online trials, subjects can adjust to decoder perturbations that are within the same intrinsic manifold (Sadtler et al., 2014). We found that individual channel activations change on up to 61.7% of channels that are important for decoding movements (Table I), and this introduces error into model predictions. However, if the perturbations we introduced did not shift activity outside of the intrinsic manifold, then it may have been easy to adjust to the new context. Our data show a near instantaneous adaptation to the perturbations whereas Sadtler et al. found some within-manifold perturbations required on the order of hundreds of trials to adapt to, indicating that our perturbation was intuitive to adapt to. Analyzing the monkey’s strategy during the BMI task revealed that they were able to do the BMI task with perturbations to the hand without adjusting their strategy (Figure 5D). This likely follows from the BMI task being driven by neural activity and visual feedback rather than movements of the hand itself. In the two decoder BMI task where off-context performance was often slightly worse, the monkeys did make small adjustments to perform the task (Figure 6D). For example, they tended to “push” harder to flex the virtual hand during off-context trials. This reaiming strategy is similar to what has been described in other work on short-term learning with motor BMI (Golub et al., 2018; Jarosiewicz et al., 2008). In this case re-aiming likely stems from the need to reproduce a higher or lower neural activation online in order to use a model trained on data where channel activation increased for flexion and decreased for extension for the spring context (Figure 7B-C).”

3. The authors should provide a clearer description of how the present findings, which are in a very limited and narrow set of conditions, further BMI design principles in general. That is, how do the authors think the present results acquired in a limited context will generalize to the much wider set of contextual shifts that would be encountered in the real world.

We thank the reviewers for the suggestion. We have added text to relate the context changes to the wider range of shifts experienced in activities of daily living.

Introduction starting line 107:

“These context shifts represent a small range of the possible shifts but relate back to common musculoskeletal changes in the task, i.e. muscle length and activation.”

Discussion starting line 438:

“These context changes represent a small sample of alterations found in activities of daily living but they include common changes to musculoskeletal properties of the hand during the task such as muscle tendon length and muscle activation range that allow us to infer how the results would extend to a wider range of changes.”

Additionally, we have expanded the discussion on how these findings further BMI design principles as well as rewording the current discussion to emphasize how the results affect BMI design. Discussion starting line 500:

“The online BMI experiments in this study used a kinematic-based BMI decoder. BMI studies typically predict kinematic variables for applications such as prosthesis control (Hochberg et al., 2012; Wodlinger et al., 2015) and cursor or virtual movement control (Gilja et al., 2015; Hochberg et al., 2006; Young et al., 2019). In the offline predictions using linear models, we found that neither kinematics nor muscle activations could be predicted at the same accuracy in new contexts. While significant, kinematics, specifically flexion velocity, did show a smaller decrease in offline performance between contexts (Figure 4B-C). These results suggest that when designing BMI, using kinematic variables as a command signal may allow for better generalization when the biomechanics of the task are not important, such as virtual tasks.

However, in FES applications (Ajiboye et al., 2017; Bouton et al., 2016; Nason-Tomaszewski et al., 2022), biomechanics are important. The final outputs are stimulation parameters that cause a desired amount of muscle contraction. Importantly, the required stimulation parameters could change with context due to the change in required muscle activation. As a result, even if predictions of position or velocity generalize well to new contexts, the mapping from kinematics to stimulation parameters would no longer be accurate. Our results with online BMI indicate that the monkeys are able to adapt by re-aiming with the BMI to restore some ability to do the virtual online task, which indicates they may also be able to re-aim in FES applications as well. However, in our task this adaptation occurred with a performance loss (Figure 6B). Instead it would be better to account for how context changes the biomechanics of the task with the BMI. This could be done either through incorporating a better control system into the BMI, e.g. developing a controller to update stimulation parameters to match the decoded joint angle or velocity, or by better estimating the intended muscle activations from neural activity. Decoded intended muscle activations can be mapped to stimulation parameters as done by some FES studies (Ethier et al., 2012; Hasse et al., 2022).”

Also discussion paragraph starting line 553 on how to make better predictions:

“An alternative BMI design approach to decoding movements is to use task-specific features to augment decoder models (Schroeder et al., 2022). The context shifts studied here represent a small and discrete subset of the shifts found in activities of daily living, however they relate to continuous musculoskeletal properties that are shifting with the context, i.e. muscle length, co-contraction, or muscle activation magnitude. Identifying a feature in neural activity that accounts for the change in muscle activation across contexts would assist in decoder generalization. For example, the contextdependent neural activity that strongly correlated with muscle activations in new contexts (Figure 8C) could provide a feature for accounting for the scaling change while predicting muscle activation or allow models to modulate force or muscle activation while producing the same kinematics. More work is needed to understand if a neural feature like this would remain stable in different cognitive contexts, for example grasping or freely moving fingers as opposed to doing this virtual target acquisition task.”

4. There are a number of typos and general quality concerns with respect to the figures and the associated captions that are at the level that negatively influence the interpretability of the study and its findings. Specific suggestions are provided by the individual reviewers but a thorough revision is required in this respect.

A thorough revision has been done, checking for typos and grammar as well as editing figures and fonts for clarity.

Reviewer #1 (Recommendations for the authors):A figure with pictures or drawings of the manipulandum in different context would be useful to better understand the experiment. Same for the virtual task/virtual hand. I'm not sure what the monkey.

We thank the reviewer for pointing this out. As described in common response #1, we have included a new Figure 1 that better illustrates the experimental setup.

Perhaps the main body of the manuscript should contain a few words about what it means for channels to be 'tuned'. What criterion? Same for the definition of "changing activity".

We thank the reviewer for the suggestion. We have added the following sentences in the Results section line 177, before table I, briefly explaining how tuning and activity changes are determined:

“Tuning and context modulation were determined by regressing finger kinematics with channel activity and channel activity multiplied by a dummy variable for context, as described in the methods, one channel at a time. Regression coefficients were tested for significance with a t-test, a significant channel activity coefficient indicated that channel was tuned, and a significant dummy variable coefficient indicated that context modulated the channel’s tuning.”

On line 205 and Figure 2: if the spring resist flexion, why does the monkey activate is extensor muscle during flexion? To resist flexing too much? In this condition, is there tonic extensor muscle activity at the beginning of the trials? Moreover, this change in extensor and flexor synergy means that the monkeys not only scale their muscle activity for a different load, but seem to change their motor strategy. The different contexts thus require a non-linear change in muscle activity, but require more or less the same kinematics. How this impacts the decoders ability to generalize should be discussed.

This is an excellent discussion point. We believe that the extensor muscle activation during flexion reflects the monkey’s strategy of moving more stiffly (i.e. co-contracting flexor and extensor muscles) in order to make controlled movements. We have added text noting this effect for Figure 2 and Figure 3 (the old Figure 1 and Figure 2).

Results line 165:

“Interestingly, even extensor muscles were more activated during spring flexion trials, indicating that monkey N was co-contracting muscles more and moving with more stiffness.”

Results line 228:

“Alternatively, during flexion, the off-context predictions of EDC activation are less correlated with measured EDC activation because the model predicts EDC inactivation, which occurs during normal trials. However, in spring trials, measured EDC activation actually increases for flexion due to Monkey N co-contracting to move more stiffly.”

This is also now discussed in the Discussion section, starting line 539:

“The lower correlation was partially driven by muscle activation patterns not observed in normal context training data, such as increased co-contracting flexor and extensor muscles during flexion trials to modulate stiffness when springs were present, as seen in Figure 3A where the predicted EDC activation does not increase for flexion in the spring condition. With a better model, it might be possible to pick out the relationship between neural activity and muscle activations. Determining for example that the intention is to activate EDC more, whether that is to co-contract with FDP or to extend the fingers may not matter as long as the intention can be accurately decoded. Based on these results, it’s likely that linear models are not able to pick out this relationship.”

Line 252-253: "This overall lack of change was surprising…" I fail to see how changing the hand configuration should impact online BMI performance. Because of differences in afferent, proprioceptive activity reaching the motor cortex? If the task is purely virtual, and the decoder good, the monkey can simply continue to try doing the "normal" task, rely on virtual feedback and should succeed. The "plant" in this case, does not change. It would be good to clarify if and how this was really an unexpected result.

This is a good question. The text overstates the surprise at the experimental results. We expected that if we were decoding from motor potent neural activity then the monkey would move their hand with the task and that the effort to move their fingers after the context change would affect the ability to do the BMI task. However, we have also seen that the monkeys can do the task with their hands restrained relying on virtual feedback. We have reworded that sentence now on line 284:

“This overall lack of change was somewhat surprising since the offline decoding results had greater prediction error. The expectation was that when the monkey moved their hand along with the BMI task, the performance would be impacted due to the context change. However, the data show that the monkeys made small adjustments to how their hand moved with the online task (Figure 5 Supplement 1).”

Line 312: it would be helpful if the "window" was described here within the main manuscript

Thank you for pointing this out, we clarified in this line (now line 383) that the window is 420ms around peak movement.

Figure 7: make it clear in the axes labels of panel C that "component 1 activation" refers to the first context-dependent component, not the first PC.

Thank you for the suggestion. We have changed the axes label to “Context-Dependent Component 1 Activation”.

Methods line 561: indicate that the rotation is the direction of wrist flexion.

Thank you for the suggestion. We have clarified in the Methods, line 688, that the wrist rotation is towards flexion:

“The second alteration was the rotation of the manipulandum by 23 degrees in the flexion direction, referred to as the “wrist context”*.”*

Reviewer #2 (Recommendations for the authors):1. The authors state that BMI performance was minimally affected by context shifts because the animals were able to make fast adjustments online. However, they do not really dig into this adjustment. I recognize that there are only a few adjustment trials per context shift, but given the number of days and shifts on some of those days, I would imagine one could still examine the process of this adaptation in some detail. There are some nice examples of this in the literature, for example Golub et al. 2018 "Learning by neural reassociation" and Athalye et al. 2018. "Emergence of Coordinated Neural Dynamics Underlies Neuroprosthetic Learning and Skillful Control".This seems like a missed opportunity to shed light on what cognitive strategies and/or neural tuning shifts. That in turn would lead to a more satisfying deeper story, which right now is essentially "neural activity changes between contexts, but BMI performance is okay, whew!".

Thank you for the specific suggestions. We have now implemented a re-aiming analysis similar to that done in Golub et al., 2018. The specific changes are detailed in the common response #2 and response to essential revision #1 and #2 at the beginning of this document. To summarize them, we find that there are some small adjustments during the online task although the effect isn’t present with every session. It’s possible that not every session required large adjustments as the Kalman filters are highly similar in their velocity predictions and the BMI task is velocity controlled (now included in the supplement). Example results with this reaiming analysis are included in the online results figures (Figure 5 and Figure 6), and an analysis including all sessions is included in the supplement. We added new discussion around these results and the implications.

2. As I alluded to in my summary statement, the difference in results between Figure 5A and 5B are quite substantial; the decoder trained off-context was quite a bit worse, and my view is that the current writing does not bring this finding to sufficient attention (e.g., the next paragraph starts with "To help explain how the monkeys were able to adjust to different contexts so well…"). A convincing explanation of why the two BMI experiments gave different results was not provided, but would be helpful for establishing confidence in the key conclusion. Rather, there's a greater focus on the lack of detrimental effect in figure 5A. To further explain my perspective: if this was a decoder innovation paper, a 32.6% improvement in times to target would be a big deal! So the performance decrease observed (and the asymmetry in what decoder+context mismatch is or is not compensated for) seems noteworthy. The Discussion should address this result, compared to the current "In both cases, the monkeys adjusted for the new context as quickly as they adjusted to normal online BMI trials."

Thank you for the feedback. As part of the re-aiming analysis detailed in common response #2 and essential revision #1 and #2, we also discuss why the results were different in the two types of BMI trials.

Additionally, we now highlight the different results more throughout the results and discussion.

For example, line 346 in the Results:

“As the off-context online performance was worse in many of the two-decoder BMI sessions, we next asked if this BMI task required more adaptation than when context changes were added to the manipulandum.”

Line 366 in the Results:

“To help explain how the monkeys were able to adjust to different contexts during the online task, we further examined changes in neural activity during the offline task in different contexts.”

Line 444 in the Discussion:

“In online trials using a kinematic based BMI, the monkeys were able to quickly adjust for context changes and achieved comparable performance to normal online trials. We tested this in two ways. First, we added context changes to the manipulandum during online trials (Figure 5), which resulted in almost no change in online performance. Second, we trained two decoders (one on normal trials and the other on off-context trials) and swapped between them for closed-loop control (Figure 6), which resulted in small but significant decreases in online performance for the model trained on off-context trials.”

Line 489 in the Discussion:

“Analyzing the monkey’s strategy during the BMI task revealed that they were able to do the BMI task with perturbations to the hand without adjusting their strategy (Figure 5D). This likely follows from the BMI task being driven by neural activity and visual feedback rather than movements of the hand itself. In the two decoder BMI task where off-context performance was often slightly worse, the monkeys did make small adjustments to perform the task (Figure 6D). For example, they tended to “push” harder to flex the virtual hand during off-context trials.”

3. Figure 1C – can the authors comment on why they think there are rather different changes in peak velocity between the two monkeys in response to the same experimental manipulations (e.g., opposite sign effect for flexion in the wrist condition)?

These different changes in peak velocity suggest that the monkeys had slightly different approaches to do the task when the wrist was flexed. Monkey N moved a little more stiffly and slowly whereas Monkey W moved more quickly to reach the targets and had to correct more after the fact. We added text to the results starting at line 147 to point out these behavioral differences:

“Each monkey had slight behavioral differences in how quickly they performed the task with context changes leading to small changes in peak velocities.”

4. Lines 259-260 "This indicates that during the off-context online trials they adjusted and moved their hand differently to account for the context change". Couldn't it also be that they tried to move their hand the same as in the normal context (i.e., send the same descending motor command, with correspondingly the same firing rate patterns that the decoder picks up on and decodes "normally"), but the mechanical perturbation (e.g., springs) resulted in different hand movements. Said another way: imagine the (hypothetical, thought-experiment) scenario where there's no sensory feedback and the monkey is doing this all open-loop using whatever strategy they learned for the normal condition BCI. How are your Figure 4D results different from what we'd expect in this scenario?

This is an excellent point. In our additional analyses described in common response #2 and essential revisions #1 and #2, we found that it appears that very similar neural activity is being produced during different context trials. This suggests that the monkey is likely attempting to make the same movements in the closed loop decode but the changes added to the manipulandum are altering the trajectory of their hand. This similar neural activity is now discussed rather than the hand movements so this line has been removed.

5. Line 371-372: "In both cases, the monkeys adjusted for the new context as quickly as they adjusted to normal online BMI trials". Where was this shown? This sounds like a quantitative statement about how quickly performance reached the same level; but in the two-model (Figure 5b) tests off-context performance was substantially worse. And, I don't think the rate of adaption to normal BCI blocks was reported anywhere? Perhaps the authors meant something else and can rephrase to clarify?

This line is referring to the average acquisition time in the first 5 trials for each type of trials that is reported in Figure 6C in order to illustrate that the initial difficulty in normal and off-context trials is similar. We agree that somewhat misrepresented the analysis and the line has been removed.

6. Results are mostly consistent across both monkeys, but not every experiment or analysis is available from both muscles (e.g., EMG was only recorded from monkey N). While all the necessary information is present in the manuscript (e.g., via tables, figure legends, methods), I think the manuscript could be more clear in the main text to clarify which results are one-monkey results and which are two-monkey results. This helps the reader better assess the strength of evidence for each finding.

Thank you for the suggestion. We have now added wording in multiple places calling out whether a result was with one or both monkeys. For example, results line 206:

“We first present the results for decoding Monkey N’s muscle activations across contexts.”

Reviewer #3 (Recommendations for the authors):I have a number of specific.1) It was not very easy to visualize what exactly the manipulandum was or what the effects of the springs were. A methods figure, perhaps as supplemental information would be useful. Related to this, it was not clear what the actual biomechanical effect of the springs were. Did they resist motion at specific joints, or across the entire digit. Can you also provide some details on the relative scale of the change in force (9.5N) to the range of grasp forces that normally be applied by the animal, or that would typically be applied in this task?

Thank you for the suggestion, we have included a new Figure 1 which illustrates the task setup as illustrated in common response #1.

Spring strengths were chosen to make the task difficult but still achievable. To give a scale for the amount of change this was to the task, text was added in the methods to give the typical force required in the task, starting line 696:

“The added springs increased the force required for full flexion by 9.5N (for MRS) and 3.3N (for index), while the rubber bands increased the force required for full flexion by 16.5N. The rubber band context was only done by Monkey N and in a 1-DOF task due to task difficulty. For reference, full flexion required approximately 1.3N of force without the springs or bands.”

2) Based on the comment in the public review about whether M1 represents or generates movements, I would make two potential suggestions. You could simplify/reduce this text and streamline the introduction, or you could weave this idea and narrative throughout the manuscript, returning to it in the discussion. As it stands, this potentially important idea is raised, but then largely ignored for the rest of the paper.

Thank you for the comment, as detailed in the response to the public review, we have written a new paragraph relating these results to the concept of generating movement.

3) In Figure 1D, what is the explanation for significantly increased activity in finger extensors during flexion in the spring context? Is this evidence of increased digit stiffness? If so, this could have interesting effects on population activity, as there is some evidence that stiffness control might be encoded in the cortex.

This is an excellent discussion point. We believe that the extensor muscle activation during flexion reflects the monkey’s strategy of moving more stiffly (i.e. co-contracting flexor and extensor muscles) in order to make controlled movements. We have added text noting this effect for Figure 2 and Figure 3.

Results line 165:

“Interestingly, even extensor muscles were more activated during spring flexion trials, indicating that monkey N was co-contracting muscles more and moving with more stiffness.”

Results line 228:

“Alternatively, during flexion, the off-context predictions of EDC activation are less correlated with measured EDC activation because the model predicts EDC inactivation, which occurs during normal trials. However, in spring trials, measured EDC activation actually increases for flexion due to Monkey N co-contracting to move more stiffly.

This is also now discussed in the Discussion section, starting line 539:

“The lower correlation was partially driven by muscle activation patterns not observed in normal context training data, such as increased co-contracting flexor and extensor muscles during flexion trials to modulate stiffness when springs were present, as seen in Figure 3A where the predicted EDC activation does not increase for flexion in the spring condition. With a better model, it might be possible to pick out the relationship between neural activity and muscle activations. Determining for example that the intention is to activate EDC more, whether that is to co-contract with FDP or to extend the fingers may not matter as long as the intention can be accurately decoded. Based on these results, it’s likely that linear models are not able to pick out this relationship.”

4) The method of calculating the Acquisition Time was not clear. Specifically, please provide a clearer description of how each dot in figure 4C and 5C are calculated. In the text description of 4C and 5C, it was difficult to interpret the text regarding which differences were significant and which were not, and what the differences in the text were referring to. For example, in Lines 263-266, it states that Monkey N had a short adaptation time (p = 0.002) and that Monkey W did not (p=0.22). I can't tell what exactly this is referring to in Figure 4C. Lastly, around line 290-292, it is stated that monkey W had no significant adaptation period. While I understand that the median z-score was close to 0, the range of z-scores in all cases in Figure 4C and 5C are very large. Were these randomly distributed across time? Was there a pattern? Can you say anything at all about the range of the z-scores themselves?

Thank you for the suggestions. A description of acquisition time and the z-scoring method has been added to the results the first time it is introduced. Additionally, we have clarified the text describing Figures 4C and 5C which previously worded the differences in z-scored acquisition times as time to adapt. The analysis was aimed at quantifying how difficult the BMI was to use initially. Results section starting line 289:

To measure the amount that the monkeys had to adjust during online trials to get to average performance, we calculated the average acquisition time, defined as the time to reach the target plus the time to finish orbiting the target, for the first five trials after the start of online trials and compared that between normal and off-context runs of BMI trials.

Also the Results section starting line 341:

“Monkey W once again did not show a significant initial adaptation (p=0.17, one-sample KolmogorovSmirnov test) which was the same between using the normal model and off-context models (p=0.5, two-sample Kolmogorov-Smirnov test).”

5) In the text, Figures 4D and 5D are discussed before 4C and 5C. Consider reorganizing the text or figure layout.

Figures 4D and 5D have been replaced (4D moved to the supplement) and the new text discussing these Figures was placed after the text discussing Figures 4C and 5C.

6) Regarding Figures 4D and 5D, is it possible to examine the distributions of the correlations in the normal and off-context trials and show this as error bars on each point? Can you statistically test whether the correlations really were different from each other to help support the claims in lines 280-290?

Figure 5D has been replaced and Figure 4D has been moved to the supplement.

7) Throughout the discussion, it would be helpful to refer back to specific figures to support claims about the results of the manuscript.

Thank you for the suggestion, multiple references to figures have been added to the discussion, for example discussion starting line 452:

“During the offline tasks, many channels changed neural activity with context, with 20.9% to 61.7% of tuned SBP channels modulating activity with context (Table I). The magnitude of these shifts were relatively small, especially when compared to the large changes in required muscle activation (Figure 2D-E), with weak trends to require greater activation for resisted flexion and lesser for assisted extension (Figure 7B-C). Additionally, the neural manifolds underlying movements in each context were well-aligned (Figure 7D). Using dPCA we found that while a large proportion of neural variance was explained by dPCA components that did not change with context, a significant proportion of the neural variance is associated with components that are context-dependent (Figure 8B). Visually, the context components are shifting the trajectories without changing the overall shape and the shift in neural activity is strongly correlated with muscle activations in new contexts (Figure 8C). This agrees with other studies which found lower variance activity may be related to the actual motor commands (Gallego et al., 2018; Russo et al., 2018; Saxena et al., 2022).”

8) The description of SBP data processing is unclear around Lines 497 and 504. What does it mean to sum the samples and track the quantity of samples? Please elaborate on what was done.

The purpose of summing samples and tracking the quantity has been elaborated in the Methods, starting line 620:

“The filtered 2kHz recording was then sent to the computer running xPC Target, which rectified and summed the samples on each channel received in each 1ms iteration and counted the quantity of samples received each 1ms so that SBP could later be averaged within longer time bins.”

9) EMG channel differencing is unclear. On Lines 512-514, it sounds like you just did the difference of the 2 channels, which is fine. Was this done digitally after the fact? Please explain. This may help clarify the statement about also recording the 'partner' electrode.

This was a software reference implemented on the analog input via Blackrock Microsystems Central software. The reference to partner electrodes has been removed as they were not used in analysis. Starting Methods line 634:

“The CPD was configured to record 16 channels of raw signal at 10 kHz and for each bipolar pair the electrode implanted further inside the muscle was software referenced to the second electrode. These eight bipolar referenced channels are used in analyses.”

10) Line 517: Provide additional details on the bandpass filter. What was the type and order of the filter?

Additional details about the filter have been added, Methods line 639:

“For offline analysis, muscle activations are estimated from the 10kHz EMG recording by filtering with a second-order Butterworth bandpass filter between 100 and 500 Hz and then taking the mean absolute value of the filtered signal during every binning period.”

11) Line 645: How common were unsuccessful trials?

Added Methods line 732:

“Unsuccessful trials were rare, often only occurring on the first or last trial of a block of trials.”